# Spontaneous crystallization of strongly confined $CsSn_xPb_{1-x}I_3$ perovskite colloidal quantum dots at room temperature

Louwen Zhang [1,2,3], Hai Zhou [1] ✉, Yibo Chen[4], Zhimiao Zheng[2], Lishuai Huang[2], Chen Wang[2], Kailian Dong[2], Zhongqiang Hu [3], Weijun Ke [2] & Guojia Fang [2] ✉

The scalable and low-cost room temperature (RT) synthesis for pure-iodine all-inorganic perovskite colloidal quantum dots (QDs) is a challenge due to the phase transition induced by thermal unequilibrium. Here, we introduce a direct RT strongly confined spontaneous crystallization strategy in a Cs-deficient reaction system without polar solvents for synthesizing stable pure-iodine all-inorganic tin-lead (Sn-Pb) alloyed perovskite colloidal QDs, which exhibit bright yellow luminescence. By tuning the ratio of Cs/Pb precursors, the size confinement effect and optical band gap of the resultant $CsSn_xPb_{1-x}I_3$ perovskite QDs can be well controlled. This strongly confined RT approach is universal for wider bandgap bromine- and chlorine-based all-inorganic and iodine-based hybrid perovskite QDs. The alloyed $CsSn_{0.09}Pb_{0.91}I_3$ QDs show superior yellow emission properties with prolonged carrier lifetime and significantly increased colloidal stability compared to the pristine $CsPbI_3$ QDs, which is enabled by strong size confinement, $Sn^{2+}$ passivation and enhanced formation energy. These findings provide a RT size-stabilized synthesis pathway to achieve high-performance pure-iodine all-inorganic Sn-Pb mixed perovskite colloidal QDs for optoelectronic applications.

Currently, all-inorganic metal halide $ABX_3$ perovskite (where A = Cs; B = Pb, Sn; and X = Cl, Br, I) nanomaterials are considered as next-generation optoelectronic functional materials with great application prospects due to their excellent photophysical properties and low-cost preparation[1–5]. In particular, the synthesis and application of cesium lead halide perovskite ($CsPbX_3$) colloidal quantum dots (QDs)/nanocrystals (NCs) have made significant progresses in optoelectronic fields such as light-emitting diodes (LEDs), solar cells, and photodetectors[6–12]. Despite the rapid development of the versatile application for perovskite colloidal QDs, the synthesis of perovskite QD colloids has always been a focus of attention for numerous researchers[13–19]. Up to now, various preparation techniques were

developed for metal halide perovskite colloidal QDs, among which the most commonly methods are the conventional hot-injection (HI) and room temperature (RT) ligand-assisted reprecipitation (LARP) technique[5,6,14]. Typically, perovskite colloidal QDs are synthesized with high crystallization quality through the HI method[13], which relies on inert gas atmosphere and high reaction temperatures (typically ≈180 °C). However, this approach is not suitable for low-cost large-scale production and severely limits its further commercial applications. In contrast, RT synthesis without inert gas protection has attracted more attention due to its easy manipulation and mild reaction conditions[15].

Previous studies have shown that the non-polar solvent-induced reprecipitation at RT is an efficient approach to preparing

[1]International School of Microelectronics, Dongguan University of Technology, Dongguan 523808 Guangdong, P. R. China. [2]Key Lab of Artifcial Micro- and Nano-Structures of Ministry of Education of China, School of Physics and Technology, Wuhan University, Wuhan 430072, P. R. China. [3]School of Electronic Science and Engineering, Xi'an Jiaotong University, Xi'an 710049, P. R. China. [4]Institute of Fluid Physics, China Academy of Engineering Physics, Mianyang 621900, P. R. China. ✉e-mail: hizhou@dgut.edu.cn; gjfang@whu.edu.cn

green-emissive lead bromide perovskite colloidal QDs similarly with HI method. However, RT synthesis in open air is an enormous challenge for pure-iodine perovskite CsPbI$_3$ QDs, mainly due to the phase transition from metastable photoactive perovskite phase to stable non-perovskite yellow phase induced by thermal un-equilibrium[20-22]. However, the RT stability of perovskite-phase iodine-based QDs can be highly improved by the reduced QD size and increased surface Gibbs energy[9,23]. It is worth noting that the polar solvents (e.g., dimethylformamide, DMF) used to dissolve precursor powders are not conducive to the formation of small-sized pure-iodine perovskite QDs and can cause damage to the crystal structure. In addition, RT synthesis of pure-iodine Cs-based all-inorganic perovskite QDs shows greater difficulty compared to organic-inorganic hybrid counterparts, because the ionic size of Cs$^+$ is not as large as that of CH(NH$_2$)$_2^+$ (FA$^+$) or CH$_3$NH$_3^+$ (MA$^+$) organic A-site cations, which is not enough to maintain the lead-iodine octahedron to form a periodic perovskite structure, resulting in the undesirable crystal distortion and structural transformation[24]. Thus, it is imperative to develop a RT reaction strategy without organic polar solvents for size-stabilized pure-iodine all-inorganic perovskite QDs.

Lead iodide perovskite QDs are usually prepared by HI method and widely available for the fields of luminescence and solar cells. The doping/alloying of B-site divalent ions (such as Mn$^{2+}$, Sr$^{2+}$, and Ca$^{2+}$) achieved by the HI method is an effective strategy for defect passivation and stability improvement of iodine-based perovskite QDs[25-32]. However, to our knowledge, the studies on direct and rapid RT alloying of CsPbI$_3$ QDs with CsSnI$_3$ in open air have never been reported before. Moreover, it is difficult to prepare stable sub-5 nm sized iodine-based perovskite QDs because of the soft ionic nature and poor crystal quality, resulting from significantly increased surface defects. Therefore, the RT preparation of small-sized and stable pure-iodine all-inorganic Sn-Pb alloyed perovskite QDs is worth further exploration, and the underlying mechanisms behind the superior properties upon RT alloying also need to be clearly elaborated.

In this work, a direct and rapid RT strongly confined spontaneous crystallization strategy in open air is proposed for preparing sub-5 nm sized pure-iodine all-inorganic CsSn$_x$Pb$_{1-x}$I$_3$ perovskite colloidal QDs with bright yellow luminescence. The experimental and theoretical studies both confirm that the yellow emission can be ascribed to the increased optical bandgap caused by strong size confinement. This study showcases that the yellow emission with such a wider bandgap is presented for iodine-based all-inorganic Sn-Pb alloyed perovskite QDs. Most notably, the size confinement and photoluminescence (PL) emission of the resultant perovskite QDs can be regulated, which benefits from the management in Cs/Pb feed molar ratio. The optical bandgaps of bromine- and chlorine-based all-inorganic Sn-Pb perovskite QDs also show a increase compared with those in the reported works. This RT synthesis strategy is also applicable to pure-iodine organic-inorganic hybrid perovskite QDs. In addition, compared with the pristine CsPbI$_3$ QDs synthesized using this RT method, the pure-iodine all-inorganic CsSn$_{0.09}$Pb$_{0.91}$I$_3$ perovskite QDs show the similar PL characteristics with superior emission properties and colloidal stability, resulting from size-stabilized crystallization and direct RT alloying of stannous ions.

## Results

### RT synthesis of CsSn$_x$Pb$_{1-x}$I$_3$ perovskite colloidal QDs

Pure-iodine all-inorganic CsSn$_x$Pb$_{1-x}$I$_3$ mixed-cation perovskite colloidal QDs with a wide bandgap were synthesized using a RT reaction method at open air via strongly confined spontaneous crystallization strategy without the involvement of organic polar solvents (see Methods), as shown in Fig. 1a. To avoid the destruction of polar solvents to the perovskite crystal structure during the reaction process, we adopted acetate salts as lead and cesium precursors, which were easily soluble in oleic acid (OA) solution. Additionally, both tin and iodine precursors were in a liquid form. Therefore, these constituent elements could undergo RT spontaneous crystallization reaction with the assistance of surfactants in non-polar toluene solvent, providing favorable formation conditions for the preparation of all-inorganic

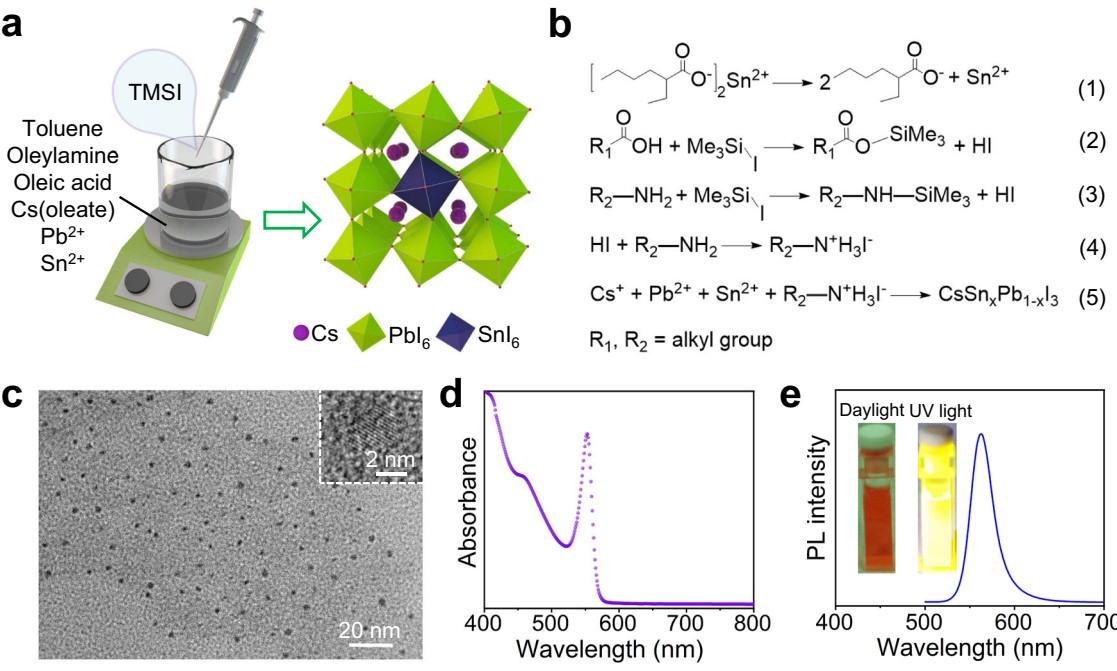

**Fig. 1 | Synthesis and characterization of CsSn$_{0.09}$Pb$_{0.91}$I$_3$ QDs. a** Schematic illustration for RT strongly confined colloidal synthesis and **b** the corresponding reaction mechanism of pure-iodine all-inorganic Sn-Pb alloyed perovskite QDs. **c** TEM image of CsSn$_{0.09}$Pb$_{0.91}$I$_3$ QDs. Inset of **c**: the corresponding HRTEM image. **d** Steady-state absorption and **e** PL spectrum (FWHM: 29 nm) of CsSn$_{0.09}$Pb$_{0.91}$I$_3$ QDs. The insets of **e**: photographs of the QD colloidal solution under daylight and UV light, respectively. Source data are provided as a Source Data file.

pure-iodine perovskite QDs with soft ionic nature (Fig. 1a). Particularly, the stable $CsSn_xPb_{1-x}I_3$ colloidal QDs with bright yellow emission were unexpectedly prepared in a Cs-deficient reaction system corresponding to Cs/Pb precursor ratio of 1:4, which was the core discovery of this work enabled by size-stabilized perovskite phase structure and stannous alloying. When trimethylsilyl iodide (TMSI) solution was added into the solution containing precursor-ligand complexes, a blood-red colloidal solution appeared immediately and emitted bright yellow luminescence under 365 nm ultraviolet (UV) excitation, as shown in Suppl. Fig. 1. This phenomenon directly reflected the successful synthesis of the high-quality pure-iodine $CsSn_xPb_{1-x}I_3$ perovskite colloidal QDs, largely due to polar-solvent-free synthetic environment and favorable thermodynamics of iodine source (TMSI) with iodine-rich supply.

After RT stirring for several minutes, the purified $CsSn_xPb_{1-x}I_3$ colloidal QDs were further obtained through centrifugation treatment. The test result of inductively coupled plasma-optical emission spectrometry (ICP-OES) for the resultant $CsSn_xPb_{1-x}I_3$ QDs showed that the actual proportion of Sn/(Sn+Pb) in the yellow emissive sample was 9.1%, revealing the composition of the Sn-Pb alloyed perovskite QDs as $CsSn_{0.09}Pb_{0.91}I_3$. The specific reaction processes of RT synthesis for pure-iodine all-inorganic Sn-Pb alloyed perovskite QDs are illustrated in Fig. 1b. The iodine ions in TMSI were released through the formation of alkyl ammonium halides, which initiated a rapid salt metathesis reaction and resulted in the nucleation and growth of the $CsSn_xPb_{1-x}I_3$ perovskite QDs. Because the growth rate of halide perovskite QDs is extremely fast, it is quite easy for iodine-based reaction system with sufficient reaction precursors to directly form a stable non-perovskite yellow phase without the photoactive performance through RT reaction. To overcome this issue, the size of the formed crystals is controlled by managing the proportion of precursor substances, thereby enhancing the RT phase stability of pure-iodine perovskite QDs. Thus, strong size confinement may be a key factor for the successful preparation and stable existence of the pure-iodine all-inorganic perovskite QDs. In this work, we found that the sub-5 nm sized $CsSn_xPb_{1-x}I_3$ QDs with a wide bandgap could be synthetized in a Cs-deficient reaction system due to the slowed growth rate caused by the scarcity of $Cs^+$ ions, and the small QD size enabled superior RT stability of pure-iodine perovskite phase. To further tailor the size confinement and energy bandgap of the resultant $CsSn_xPb_{1-x}I_3$ QDs, the influence of Cs/Pb precursor ratio in RT reaction system on the products was systematically studied, which will be further elaborated later.

## Characterization of perovskite colloidal QDs

Firstly, the morphology and microstructure of $CsSn_{0.09}Pb_{0.91}I_3$ perovskite QDs with yellow emission were characterized by transmission electron microscopy (TEM). As can be seen from Fig. 1c, the as-prepared $CsSn_{0.09}Pb_{0.91}I_3$ QDs exhibit nearly spherical particles with low size dispersity and sub-5 nm size. High-resolution TEM (HRTEM) image (inset of Fig. 1b and Suppl. Fig. 2) shows clearly visible lattice fringes, further revealing that the $CsSn_{0.09}Pb_{0.91}I_3$ QDs possess high crystallinity. Figure 1d, e presents the optical properties of $CsSn_{0.09}Pb_{0.91}I_3$ QDs. The steady-state absorption spectrum displays a sharp exciton absorption peak at 553 nm (Fig. 1d), which is attributed to the narrow size distribution and small particle size. As displayed in Fig. 1e, the PL emission peak position of $CsSn_{0.09}Pb_{0.91}I_3$ QDs is approximately located at 563 nm, and the full width at half maxima (FWHM) is 29 nm. As shown in the inset of Fig. 1e, the purified colloidal solution of QDs is shown in red, and the bright yellow glow can be clearly observed from the colloidal solution under UV light. Compared with the previously reported emission peak positions of pure-iodine perovskite QDs, the emission peak position in this work undergoes an unexpected notable blue-shift, resulting from the strong size confinement[18,33–36].

More importantly, the yellow emission from pure-iodine all-inorganic perovskite QDs/NCs is experimentally achieved in this study. Moreover, this synthesis technology is equally effective for organic-inorganic hybrid perovskite QDs. FA-based Sn-Pb iodide hybrid perovskite QDs show a spherical morphology and yellow emission properties similar to those of all-inorganic counterparts (Suppl. Fig. 3). However, the colloidal stability of this hybrid perovskite QDs is relatively poor due to the presence of A-site organic component. Through the same synthesis pathway along with altering halogen elements (Br or Cl), the Br- and Cl-based all-inorganic Sn-Pb alloyed perovskite QDs were also successfully prepared. The resulting perovskite QDs also exhibit significant blue-shift of exciton peaks and enlarged optical bandgap compared with those in previously reported studies, as displayed in Suppl. Fig. 4. The optical bandgap of Br-based perovskite QDs shifts from the green region to the blue region, and a dramatic shift from the blue region to the UV region is achieved for Cl-based perovskite QDs. In order to further reveal the origin of yellow emission from pure-iodine perovskite QDs in this work, the energy band structure of QDs with the actual size corresponding to the experimental result is analyzed through theoretical calculation, which will be presented in detail later.

To further identify the crystal structure of the as-prepared $CsSn_{0.09}Pb_{0.91}I_3$ QDs, the pure-lead and pure-tin all-inorganic iodide perovskite QDs were prepared separately using the same RT approach. Unfortunately, the purified pure-tin perovskite QD colloidal solution cannot be obtained in open air due to the susceptibility of stannous ions to oxygen gas. On the other hand, the pure-lead $CsPbI_3$ QDs can still be synthesized smoothly via this spontaneous crystallization strategy. The resulting $CsPbI_3$ QDs show similar morphology and size distribution to $CsSn_{0.09}Pb_{0.91}I_3$ QDs (Suppl. Figs. 5 and 6). The average size of the colloidal QDs obtained in this work is around 3–4 nm. As shown in the X-ray diffraction (XRD) patterns in Fig. 2a, the pure-iodine $CsSn_{0.09}Pb_{0.91}I_3$ QDs possess the same crystal structure as $CsPbI_3$ QDs, which accords with the γ-phase perovskite crystal structure (ICSD: 264725)[37,38]. Most notably, a slight shift towards a higher angle for the diffraction peak of $CsSn_{0.09}Pb_{0.91}I_3$ QDs can be observed compared to that of $CsPbI_3$ QDs (Suppl. Fig. 7), which is derived from lattice contraction caused by the partial substitution of $Pb^{2+}$ ions with smaller sized stannous ions. This point is also proved by the decreased interplanar spacing in HRTEM images (Suppl. Fig. 8). In addition, X-ray photoelectron spectroscopy (XPS) measurement was carried out to identify the effects of the alloyed $Sn^{2+}$ on the composition and chemical states of the resulting perovskite QDs. Suppl. Fig. 9 depicts the high-resolution XPS spectra for Pb $4f$, I $3d$ and Sn $3d$, all calibrated with C 1s. The peaks slightly shift towards higher binding energy in Pb $4f$ and I $3d$ core-level spectra of the QD films alloyed with stannous ions. The two peaks located at 495.0 and 486.5 eV for $CsSn_{0.09}Pb_{0.91}I_3$ QDs are attributed to divalent Sn $3d_{3/2}$ and $3d_{5/2}$, respectively. These results further indicate that the stannous ions have successfully incorporated into the $CsPbI_3$ perovskite crystal structure and achieved partial substitution of B-site lead ions.

## Photophysical properties

We further conducted comparative studies on the optical properties of $CsPbI_3$ and $CsSn_{0.09}Pb_{0.91}I_3$ QDs. Both the exciton optical absorption and PL emission peaks almost have no significant differences for $CsPbI_3$ and $CsSn_{0.09}Pb_{0.91}I_3$ QDs, as shown in Fig. 2b, c. However, the PL emission intensity of $CsSn_{0.09}Pb_{0.91}I_3$ QDs is much higher than that of $CsPbI_3$ QDs in PL spectra, which can also be seen from the photographs of the colloidal solutions under UV light (the inset of Fig. 2c). The PL quantum yield (QY) measurement of the resultant QDs was further performed. The PL QY value (55.4%) of pure-iodine $CsSn_{0.09}Pb_{0.91}I_3$ QDs was significantly higher than that (21.7%) of $CsPbI_3$ QDs, indicating the higher emission efficiency for $CsPbI_3$ QDs alloyed with stannous

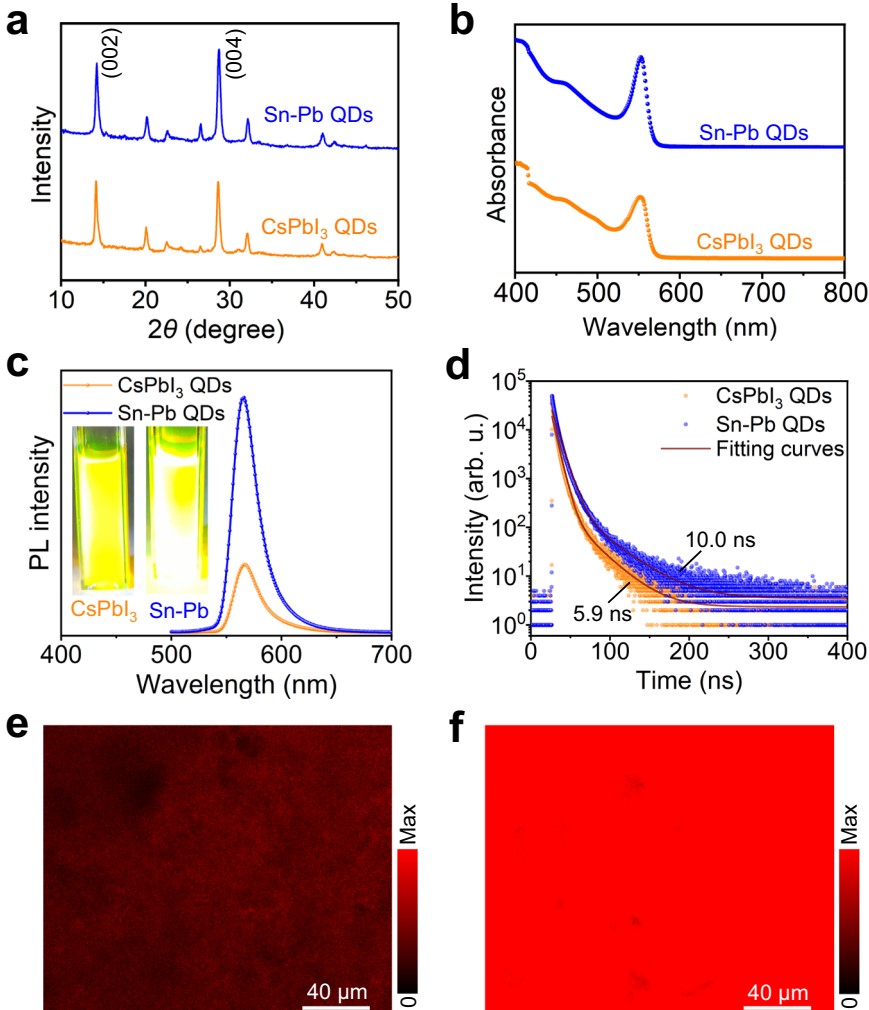

**Fig. 2 | Comparative analysis of photophysical properties for CsPbI₃ and CsSn₀.₀₉Pb₀.₉₁I₃ QDs. a** XRD patterns of CsPbI₃ and CsSn$_{0.09}$Pb$_{0.91}$I₃ QDs. **b** Steady-state absorption, **c** PL spectra and **d** PL decay curves of CsPbI₃ and CsSn$_{0.09}$Pb$_{0.91}$I₃ QDs. The inset of **c**: photographs of the colloidal solutions under UV light. PL mapping images of **e** CsPbI₃ and **f** CsSn$_{0.09}$Pb$_{0.91}$I₃ QDs dripped on glass slides. Source data are provided as a Source Data file.

ions. As shown in Fig. 2d, it is observed that the PL decay of CsPbI₃ QDs is faster than that of CsSn$_{0.09}$Pb$_{0.91}$I₃ QDs. The PL decay curves were fitted with biexponential decay functions[39], which can be divided into a fast component ($\tau_1$, corresponding to defect-induced nonradiative recombination) and a slow component ($\tau_2$, excitonic radiative recombination). The average PL lifetime ($\tau_{avg}$) is defined as:

$$\tau_{avg} = (A_1 \tau_1^2 + A_2 \tau_2^2)/(A_1 \tau_1 + A_2 \tau_2) \tag{1}$$

where $A_1$ and $A_2$ parameters are taken from the fitting results of the decay curves. The CsSn$_{0.09}$Pb$_{0.91}$I₃ QDs exhibit an extended $\tau_{avg}$ (10.0 ns) as compared to the pristine CsPbI₃ QDs (5.9 ns), which indicates that the surface defects related to nonradiative recombination are dramatically reduced when stannous ions are incorporated into the CsPbI₃ lattice. As a note, the carrier lifetime in this work is much shorter than that of CsPbI₃ QDs/NCs in the reported studies, mainly due to the large number of surface defect states caused by the small QD size. The spectroscopic results indicate that CsSn$_{0.09}$Pb$_{0.91}$I₃ QDs have superior PL properties owing to the incorporation of stannous ions. Furthermore, the PL mapping measurement of QD films on glass substrates also confirms that the PL intensity of CsSn$_{0.09}$Pb$_{0.91}$I₃ QDs is much stronger than that of CsPbI₃ QDs, as displayed in Fig. 2e, f. There are numerous Pb-related defects in the crystal lattice of small-sized CsPbI₃ QDs, which can act as nonradiative recombination centers to

cause PL quenching, leading to a significant decrease in emission intensity. Therefore, the defect passivation of CsPbI₃ QDs with stannous alloying can be realized through filling Pb vacancy defects and replacing uncoordinated Pb atoms, which greatly alleviates the trap-assisted nonradiative recombination process on the QD surface, thereby boosting the exciton recombination process and enhancing the luminous efficiency.

The effects of Pb/Sn feed molar ratio on the optical properties of pure-iodine all-inorganic Sn-Pb perovskite QDs were performed, as shown in Suppl. Fig. 10. The experimental results indicate that the exciton absorption and PL emission characteristics of the as-prepared Sn-Pb perovskite QDs show negligible changes with the change in Pb/Sn ratio. However, the lifetime $\tau_{avg}$ of photogenerated charge carriers in the perovskite QDs is gradually shortened with the decreased Pb/Sn ratio (Suppl. Fig. 10c and Suppl. Table 1). Hence one can see that the excessive incorporation of stannous ions will cause an increase in defect states of Sn-Pb perovskite QDs, leading to intensified nonradiative recombination and reduced carrier lifetime. This may be mainly attributed to the intrinsic instability of stannous ions in air. In addition, we further investigated the effect of increasing reaction temperature on the optical properties of pure-iodine Sn-Pb perovskite QDs. As displayed in Suppl. Figs. 11 and 12, the QDs synthesized at 120 and 180 °C exhibit redshift optical properties with the primary emission peak at nearly 595 nm. But even if the reaction time was prolonged

from 3 to 6 min, the main emission peak position remains almost unchanged. However, the weak peak around 560 nm has been weakened. To further study the growth ability of pure-iodine Sn-Pb perovskite QDs at a relatively high temperature, the reaction time for QDs synthesized at 120 °C was further extended to 50 min, as shown in Suppl. Fig. 13. The results show that the emission peak position of the resulting QDs is still around 600 nm and there is no redshift. The perovskite QDs prepared at a high temperature exhibit an extended fluorescence lifetime due to the reduced nonradiative recombination centers and enhanced crystal quality. Nevertheless, the promotion effect of increasing synthesis temperature and prolonging reaction time on the growth of pure-iodine Sn-Pb perovskite QDs is still limited, resulting from the limited amount of Cs precursor and the inhibition of continued growth process. The effects of Cs/Pb feed molar ratio on the resultant QDs will be further systematically studied in the following text. It is the reason that the strongly confined pure-iodine Sn-Pb perovskite QDs with the yellow emission are successfully prepared at RT in open air.

### Tunable optical bandgap of $CsSn_xPb_{1-x}I_3$ QDs

The luminescent properties of iodine-based perovskite QDs synthesized at RT in this work are completely different from the narrow-band deep-red emission reported previously[5,9]. It can be considered as a result of strong size confinement. The electronic band structures of about 3.6 nm sized $CsPbI_3$ QDs and pure-iodine Sn-Pb perovskite QDs were further examined through density function theory (DFT) calculations, and the results are shown in Fig. 3a, b. One can see from Fig. 3a that $CsPbI_3$ QDs is a direct semiconductor with theoretical bandgap value of 2.19 eV. As represented in Fig. 3b, the pure-iodine Sn-Pb perovskite QDs possess a direct bandgap of 2.12 eV, which is slightly lower than the experimental value of 2.20 eV obtained from the Tauc plot in Suppl. Fig. 14. The calculated theoretical value is generally lower than the experimental value because the fact that the Perdew-Burke-Ernzerhof (PBE) calculations may underestimate the bandgap of semiconductor materials. Besides, the confinement effect on the

optical bandgap can also be accounted for in a spherical potential well according to the following equation:

$$\triangle E_{gap} = \hbar^2 \pi^2 / (2\mu r^2) \qquad (2)$$

where $r$ is the particle radius and $\mu$ is the reduced mass of the exciton[13]. This provides favorable evidences for the size confinement effect and blue-shifted optical properties of semiconductor QDs. Therefore, the bright yellow emission in this work is attributed to direct radiative recombination corresponding to a wider bandgap caused by size confinement.

Furthermore, the pure-iodine all-inorganic Sn-Pb mixed perovskite QDs were synthesized with varying Cs/Pb feed molar ratio. To quantify these observations, PL and absorption spectra were performed on the resultant Sn-Pb perovskite colloidal QDs, as displayed in Fig. 3c. The as-obtained colloidal solutions exhibit an apparent difference in color (from blood-red to dark red, see Suppl. Fig. 15), especially under UV light, with a clear shift from yellow to red with the increased Cs/Pb ratio. The principal emission peak shows a large redshift from 560 nm for 1:4 sample to 640 nm for 3:2 sample. For UV-vis absorption measurements, the pronounced excitonic peak appears when the amount of the Cs precursor is low, whereas the sharp excitonic peak is almost invisible as the Cs/Pb ratio increases. The mechanism lies in the fact that increasing the amount of A-site Cs cations will facilitate the further growth of perovskite QDs, leading to an increase of the QD size and a spectral redshift, which is further confirmed by TEM analysis (Suppl. Fig. 16). The samples with Cs/Pb ratios of 1:1 and 1:2 exhibit high size polydispersity and different morphologies (e.g., nanoparticles, nanosheets, and nanorods), that is the reason for the multimodal nature of the PL spectra[40–42]. In general, when the amount of a reactant is relatively low, the reaction rate of the entire reaction system will slow down, thereby severely inhibiting the growth rate of QD crystals. Thus, the resultant Sn-Pb perovskite colloidal QDs shrink in size with a decreased Cs/Pb feed molar ratio, and exhibit strong size confinement. Moreover, PL decay curves of the

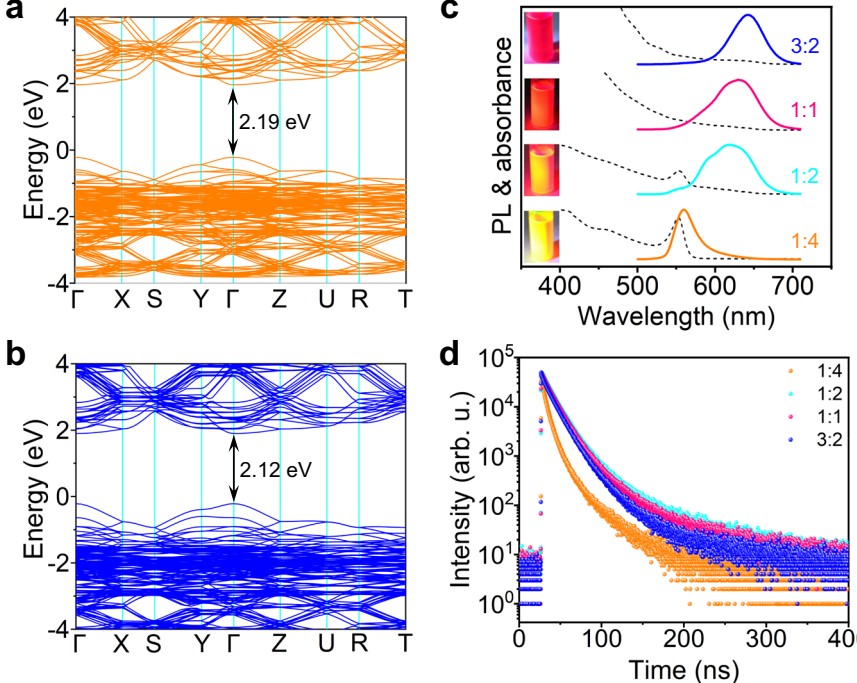

**Fig. 3 | Tunable optical bandgap enabled by varying Cs/Pb ratio.** Calculated band structure of about 3.6 nm sized **a** $CsPbI_3$ QDs and **b** $Sn^{2+}$ alloyed $CsPbI_3$ QDs. **c** PL (solid lines) and absorption (dashed lines) spectra, and **d** PL decay curves of pure-iodine all-inorganic Sn-Pb alloyed perovskite QDs with varying Cs/Pb feed molar ratio. The inset of **c**: photographs of the corresponding colloidal solutions under UV light. Source data are provided as a Source Data file.

pure-iodine Sn-Pb perovskite QDs with varying Cs/Pb feed molar ratio were recorded, as shown Fig. 3d. When the amount of Cs precursor increases, the fluorescence lifetime of the perovskite QD solution is significantly extended due to the reduced surface defects caused by the increased QD size. However, when the Cs/Pb ratio is greater than 1:2, the PL lifetime of photogenerated charge carriers is slightly shortened, possibly due to the thermodynamic instability of large-sized iodine-based perovskite crystals and the formation of a small amount of non-photoactive yellow phase in the product. As a result, the size and optical properties of the pure-iodine Sn-Pb perovskite QDs can be effectively tuned through adjusting the Cs/Pb feed ratio. It is also worth mentioning that the resultant QDs with longer emission wavelengths lose their PL performance after being placed in air for one day, and their RT stability is much lower than that of wider bandgap QDs (Suppl. Fig. 17). Therefore, the colloidal stability of the pure-iodine perovskite QDs largely benefits from the small QD size.

### The mechanism for performance enhancement

The colloidal stability of $CsSn_{0.09}Pb_{0.91}I_3$ QDs showed a great improvement in comparison with that of the pristine $CsPbI_3$ QDs. Suppl. Fig. 18 illustrates the colloidal stability of the diluted QD solutions against water (2 mL of hexane NCs solution and 1 mL of water were added in a cuvette). Pseudo-color maps of water treatment time-dependent PL spectra are presented in Fig. 4a, b. It can be seen that the yellow fluorescence of $CsPbI_3$ QDs becomes very weak after water treatment for 20 min, and the relative PL intensity drops to only 21%. In striking contrast, $CsSn_{0.09}Pb_{0.91}I_3$ QDs still exhibit bright yellow emission, and around 66% of the initial PL intensity is retained (Fig. 4c). Moreover, a comparison study on the thermal stability of the samples is exhibited in Fig. 4d–f. The diluted $CsSn_{0.09}Pb_{0.91}I_3$ QDs retain 37% of the initial PL intensity as the temperature increases from 20 to 100 °C, while only 13% of the initial intensity is maintained for the pristine $CsPbI_3$ QDs. As demonstrated in Suppl. Fig. 19, the diluted $CsSn_{0.09}Pb_{0.91}I_3$ QDs exhibit superior air stability to the pristine $CsPbI_3$ QDs. These results suggest that $CsSn_{0.09}Pb_{0.91}I_3$ QDs show significantly

increased colloidal stability due to the incorporation of stannous ions at the B-site of the $CsPbI_3$ lattice.

Overall, the iodine-based all-inorganic perovskite QDs with sub-5 nm size and wide bandgap have been successfully synthesized through RT spontaneous crystallization strategy, which opens a RT pathway for the synthesis of high-quality pure-iodine perovskite colloidal QDs. More importantly, the $CsSn_{0.09}Pb_{0.91}I_3$ QDs exhibit more excellent PL properties and colloidal stability as compared with $CsPbI_3$ QDs. To further elucidate the mechanism for the enhanced properties of QDs with the alloyed stannous ions, the electronic properties of $CsPbI_3$ perovskite lattice are performed by using the first-principles calculations based on DFT. Figure 5 depicts the projected density of states (DOS) and electronic charge density for the valence band maximum (VBM) and conduction band minimum (CBM), representing the hole and electron distributions. We consider two slab models that are (i) a $CsPbI_3$ slab with a removed lead atom, and (ii) a $CsPbI_3$ slab treated with a filled stannous atom. The lead vacancy defect in $CsPbI_3$ leads to the formation of a trap state within the bandgap (Fig. 5a), which can act as a nonradiative recombination center (Fig. 5b). It is also found that the valence bands of $CsPbI_3$ are mainly composed of I $5p$ orbital, whereas the conduction bands are mainly formed with Pb $6p$ orbital (Suppl. Fig. 20a). After introducing stannous atom, Sn $5p$ orbital also contributes to the conduction bands of $CsPbI_3$ apart from Pb $6p$ orbital (Suppl. Fig. 20b). Significantly, once the lead vacancy is occupied by stannous atom, the mid-gap states disappear completely, as shown in Fig. 5c. The highly delocalized VBM and CBM states are developed in a perfect perovskite lattice without defect states (Fig. 5d). Thus, our theoretical calculations demonstrate that the incorporated stannous ions not only fill into the lead vacancies but also passivate the defect states in $CsPbI_3$ lattice. As displayed in Suppl. Fig. 21, the formation energy of all-inorganic Sn-Pb iodide perovskite is much bigger than that of $CsPbI_3$, which indicates that the formation of vacancy defects gradually becomes difficult with the increased proportion of stannous ions in the pure-iodine Sn-Pb alloyed perovskite. This verifies that the pure-iodine all-inorganic Sn-Pb perovskite are

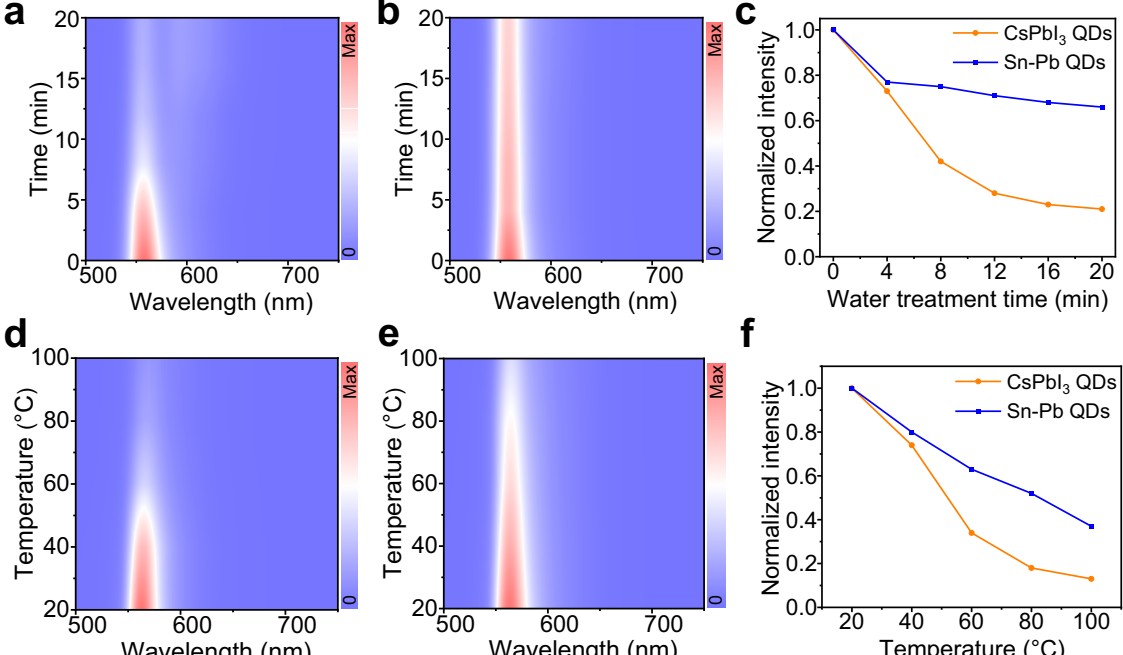

**Fig. 4 | Stability comparison of $CsPbI_3$ and $CsSn_{0.09}Pb_{0.91}I_3$ QDs against water and heat.** Pseudo-color maps of water treatment time-dependent PL spectra for **a** $CsPbI_3$ and **b** $CsSn_{0.09}Pb_{0.91}I_3$ QDs, respectively. **c** Variation in relative PL intensity of $CsPbI_3$ and $CsSn_{0.09}Pb_{0.91}I_3$ QDs with water treatment time. Pseudo-color maps of temperature-dependent PL spectra for **d** $CsPbI_3$ and **e** $CsSn_{0.09}Pb_{0.91}I_3$ QDs, respectively. **f** Variation in relative PL intensity of $CsPbI_3$ and $CsSn_{0.09}Pb_{0.91}I_3$ QDs as a function of temperature. Source data are provided as a Source Data file.

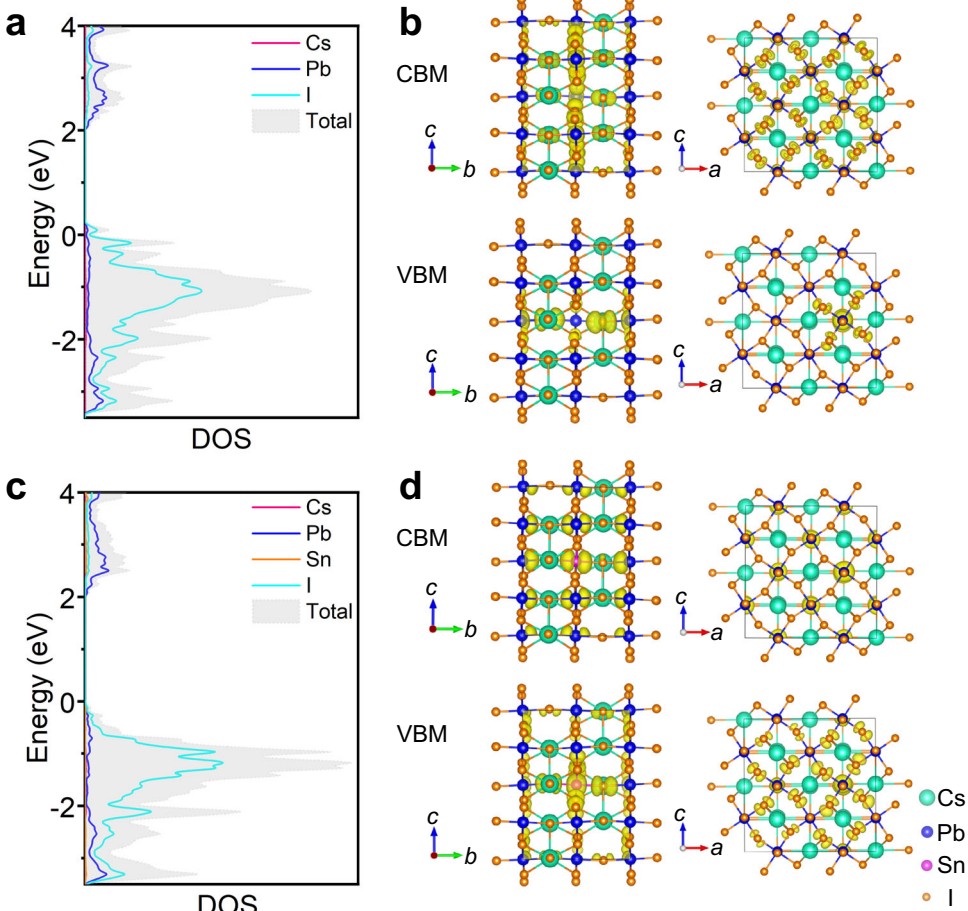

**Fig. 5 | Theoretical calculation for the role of Sn²⁺ alloyed in CsPbI₃ lattice. a, c** Projected DOS and **b, d** electronic charge density for the VBM and CBM for (**a, b**) a pristine CsPbI₃ slab with lead vacancy defects, and (**c, d**) a Sn²⁺-incorporated CsPbI₃ slab. The orange line shows the contribution from Sn²⁺ to the valence and conduction bands. The VBM is set at the position with zero energy. Source data are provided as a Source Data file.

more thermodynamically stable than the pristine CsPbI₃. Therefore, the improved properties of the pure-iodine Sn-Pb perovskite QDs are attributed to the defect passivation of stannous ions and enhanced defect formation energy.

## Discussion

In summary, the bright yellow emission from sub-5 nm CsSn₀.₀₉Pb₀.₉₁I₃ perovskite colloidal QDs was achieved through RT strongly confined spontaneous crystallization strategy without polar solvents in open air, which was attributed to direct radiative recombination corresponding to a wider bandgap derived from strong size confinement. The proposed RT synthesis of pure-iodine all-inorganic perovskite QDs was enabled by the enhanced phase stability, resulting from the small QD size. The feed ratio of Cs/Pb precursors plays a crucial role in size confinement and PL emission wavelength of the resultant QDs, while the alloying of stannous ions has negligible effect on them. However, the CsSn₀.₀₉Pb₀.₉₁I₃ QDs exhibited the enhanced PL intensity, prolonged fluorescence lifetime and improved colloidal stability as compared with the pristine CsPbI₃ QDs. The theoretical analysis based on DFT further suggested that the superior properties of CsSn₀.₀₉Pb₀.₉₁I₃ QDs were attributed to the Sn²⁺ passivation and enhanced defect formation energy. As a result, this work provides a feasible approach to prepare stable and strongly confined pure-iodine all-inorganic Sn-Pb mixed perovskite QDs under mild reaction conditions, for low-cost, highly efficient halide perovskite QDs and their optoelectronic applications.

## Methods

### Materials

Cesium acetate (C₂H₃CsO₂, 99.9%), formamidine acetate (CH₄N₂·C₂H₄O₂, 99%), tin(II) 2-ethylhexanoate (C₁₆H₃₀O₄Sn, 95%), iodotrimethylsilane (C₃H₉SiI, 97%), bromotrimethylsilane (C₃H₉SiBr, 98%), chlorotrimethylsilane (C₃H₉SiCl, >98.0%), lead(II) acetate trihydrate (PbC₄H₆O₄·3H₂O, 99.5%), OA (C₁₈H₃₄O₂, analytical reagent (AR)), oleylamine (C₁₈H₃₇N, OAm, 80–90%), n-hexane (C₆H₁₄, AR), and ethyl acetate (C₄H₈O₂, 99.5%) were purchased from Aladdin Industrial Corporation, China. Toluene (C₇H₈, AR) solvents were purchased from Sinopharm Chemical Reagent Co., Ltd. All chemicals were directly used without further purification.

### Preparation of Cs⁺ precursor solution

0.384 g cesium acetate and 6 mL OA were loaded into a 10 mL vial and stirred continuously at RT. After a while, cesium acetate powder was dissolved completely. The as-obtained precursor solution was stored at RT for the subsequent colloidal synthesis.

### Synthesis of perovskite colloidal QDs

In a typical synthesis process for CsSn₀.₀₉Pb₀.₉₁I₃ perovskite QDs, 0.152 g PbC₄H₆O₄·3H₂O, 5 mL toluene solution, 700 μL OA solution, and 300 μL Cs⁺ precursor solution were loaded into a 25 mL beaker and stirred continuously until the lead precursor was dissolved completely. Then, 500 μL OAm and 75 μL C₁₆H₃₀O₄Sn were added to the above-mentioned solution. Finally, 200 μL TMSI solution was swiftly

dropped into the mixed solution. We could immediately observe that the reaction solution turned into blood-red and emitted bright yellow fluorescence under UV light. After several minutes, the purification of QD crude solution was processed by using high-speed centrifugation. The QD crude solution was transferred into 50 mL centrifuge tubes. 15 mL ethyl acetate was added into the crude solution, and the mixture was centrifuged at $10,000\times g$ for 6 min. The precipitate was then dispersed in 20 mL hexane solution, and centrifuged at $4,500\times g$ for 3 min. The purified colloidal solution was stored at RT for further characterization. For the pristine $CsPbI_3$ perovskite QDs, except for no addition of tin precursor, other experimental details were consistent with those of $CsSn_{0.09}Pb_{0.91}I_3$ QDs mentioned above. For FA-based hybrid Sn-Pb iodide perovskite QDs, cesium acetate was replaced with formamidine acetate. For Br- or Cl-based all-inorganic Sn-Pb perovskite QDs, the iodine source was substituted by $C_3H_9SiBr$ or $C_3H_9SiCl$.

**First-principles calculations.** First-principles calculations were performed by Vienna ab-initio simulation package (VASP). The generalized gradient approximation (GGA) of PBE was used to describe the exchange-correlation functional. The cut-off energy for the plane wave basis was set to 400 eV and a $2 \times 2 \times 2$ Monkhorst-pack mesh was employed. Partial occupancies of the KohnSham orbitals were allowed using a Gaussian smearing with a width of 0.05 eV. All structures were fully relaxed (atomic position) up to $10^{-4}$ eV Å$^{-1}$ force minimization and maximum force of 0.01 eV Å$^{-1}$. To simulate quantum size, a supercell model containing 960 atoms with $a = 3.48467$ nm, $b = 3.79692$ nm and $c = 3.51517$ nm was established. For the calculation of DOS and charge distribution, a supercell model containing 80 atoms was established. Formation energies ($E_f$) of all these defects in the doped system were calculated according to Eq. (3):

$$E_f = E_{df} - E_{ini} + \sum_i n_i \mu_i \qquad (3)$$

$E_{ini}$ and $E_{df}$ are the total energies of the supercell before and after doping, respectively. $n_i$ is the number of elements added or removed during the formation of the defects, $u_i$ is the chemical potential of the element. For the metal cations, the chemical potentials were obtained from the calculations of the respective bulk materials.

**Characterization.** TEM and HRTEM images were collected by using a JEM-F200 transmission electron microscope with an acceleration voltage of 200 kV. XRD patterns of perovskite QDs were measured with a powder X-ray diffractometer (Rigku Smartlab, Japan) using Cu Kα radiation ($\lambda = 0.15418$ nm) under operating conditions of 40 kV and 44 mA. All samples for XRD testing were prepared on glass slides. XPS measurements were performed on an AXIS-ULTRA DLD-600W photoelectron spectrometer (Kratos, Japan). The actual element content of the samples was conducted by an Agilent 720ES ICP-OES. Absorption spectra were recorded by a SHIMADZU mini 1280 spectrophotometer, and PL spectra and PL decay curves were measured by a Delta Flex fluorescence spectrum spectroscopy (HORIBA). Excitation intensity for the PL decay curves was 0.03 μJ cm$^{-2}$, and the excitation wavelength was 481 nm. All tests were performed under ambient conditions at RT, unless otherwise noted.

### Reporting summary
Further information on research design is available in the Nature Portfolio Reporting Summary linked to this article.

## Data availability
The data that support the findings of this study are available from the corresponding authors upon request. Source data are provided with this paper.

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

## Acknowledgements

This work is financially supported by the National Natural Science Foundation of China (51972101, H.Z., 62074117, G.F., 12134010, G.F.), the support of the key R & D program from Hubei Province (2023BAB102, G.F.), and the Research platforms and projects of Guangdong universities in 2022 (2022ZDZX1028, H.Z.). The authors greatly appreciate the support of TEM analysis from Dr. Meimei Zhang. The authors also gratefully acknowledge Shiyanjia Lab (www.shiyanjia.com) for the ICP-OES analysis.

## Author contributions

H.Z. and G.F. supervised the research project. L.Z. conceived the idea, designed and performed the experiments. Z.Z, L.H., C.W., and K.D. assisted in conducting experiments for material characterization. Y.C., Z.H., and W.K. provided constructive suggestions for experimental analysis. L.Z., H.Z., and G.F. reviewed and edited the manuscript.

## Competing interests

The authors declare no competing interest.
