## [Peer Review File · Nature Communications]

Spontaneous crystallization of strongly confined CsSnxPb1-xI3 perovskite colloidal quantum dots at room temperatureREVIEWER COMMENTS

Reviewer #1 (Remarks to the Author):

In this manuscript, Zhang et al present a room-temperature synthesis approach for strongly confined colloidal Cs(Pb:Sn)I₃ nanocrystals (NCs). The main driving parameter for the quantum confinement seems to be the Cs/Pb precursors' ratio. While the discussed results are of interest, I have several minor and major comments that, in my opinion, should be addressed before considering publication:

1) Logical order: The tuning of the NC size by controlling Cs/Pb feed ratio is only presented in Figure 3, while Figures 1 and 2 deal already with an "optimized" ratio (1:4) whose origin the reader cannot understand at this stage. I would recommend first showing and discussing the effect of Cs/Pb ratio to get sub-5nm NCs and then focusing on the chosen ratio of 1:4. In a sense this means starting with Figure 3.

2) Figure 3c shows clear changes in PL. Ratios of 1:1 and 1:2 show broad and multi-peak signals pointing towards high size polydispersity. TEM should be included to evaluate the NC sizes (and distribution) versus precursors' ratios. Also, given the abrupt shift from 1:2 to 1:4, it may be worth exploring an intermediate ratio of 1:3.

3) The manuscript discusses results as a function of elemental feed ratios but there is no quantitative or semiquantitative analysis of the final perovskite composition. I believe this is very important to analyze the results. Keep in mind that bandgap energy of Pb-Sn perovskites is not a monotonous function of Pb:Sn ratio (bandgap bowing). Actually, the authors present XPS results, from which semi-quantitative analysis should be easy to extract.

4) In Figure S8, caption for panel (b) should be I 3d instead of Br 3d.

Reviewer #2 (Remarks to the Author):

The manuscript reported a RT strongly confined spontaneous crystallization strategy in open air for preparing sub-5 nm sized pure-iodine tin-lead (Sn-Pb) perovskite colloidal QDs with bright yellow luminescence. However, as an important part of this manuscript, this synthesis method lacks detailed description and analysis of its process. And the novelty of this work is not significant. Moreover, the analysis of some test data is insufficient. Therefore, I would recommend to reject this manuscript considering that Nature Communications is targeting high impact journal with broad readership. I think the following issues should be addressed seriously, before the next submission:

1. Detailed description and analysis of the process of the proposed "RT strongly confined spontaneous crystallization strategy" should be provided. This strategy is the key part of this paper, and more space should be devoted to discussing the reaction mechanism or some phenomena in the reaction process.
2. The authors claimed that the prepared pure-iodine tin-lead (Sn-Pb) perovskite colloidal QDs show highly bright yellow luminescence. The most fundamental and direct data of photoluminescence quantum yield should be provided to support this claim.
3. In Figure S9b, the PL spectra exhibit an obvious blueshift with the increase of the Pb/Sn ratio, which is inconsistent with the author's description that PL emission properties do not change.
4. The proof for the formation of Sn-Pb alloyed perovskite QDs is insufficient. More characterizations, such as element mapping and lattice spacing, should be provided.
5. In Figure 3c, the PL spectra are broadened and have a tendency of multiple peaks when the Cs/Pb ratio is 1:1 and 1:2. Reasonable explanations should be added.

Reviewer #3 (Remarks to the Author):

In this manuscript, the authors reported yellow emission from pure-iodine all-inorganic perovskite QDs/NCs synthesized at room temperature and observed that stannous ions can passivate the lead vacancies, leading to enhanced stability. The key point in this manuscript is the room-temperature synthesis of sub-5 nm sized pure-iodine QDs that showed peak PL at 563 nm, just as the authors claimed in the manuscript - 'yellow emission from pure-iodine QDs/NCs is experimentally achieved for the first time'.

First of all, the results in this manuscript are very similar to the reported work (Nanoscale, 2021, 13, 4899-4910). The Nanoscale paper reported uniform cubic-phase CsPbI₃ NCs with PL peak at 569 nm, which are also synthesized at room temperature in open air (i.e. same nanocrystals, same synthesis condition, and very similar yellowish emission). In addition, the Nanoscale paper also demonstrated the ultrasmall-sized CsPbBr₃ NCs with emission at 470 nm and ultrasmall-sized CsPbCl₃ NCs with emission at 385 nm, where uniform nucleation and growth and chemical stability are simultaneously achieved with their method. In addition to this, I would also like to make the authors aware of the Sn-Pb perovskite nanocrystal papers where room temperature synthesis has been developed and ultrasmall Sn-Pb dots of <3 nm has been demonstrated (ACS Energy Lett. 2017, 2, 1190-1196, Nat. Photon. 2021, 15, 696-702, Chem. Mater. 2020, 32, 1089-1100, Light Sci. Appl. 2023, 12, 208). The authors should make clear the novelty of their study, by careful comparison with the synthesis method in the papers listed above, as this idea is the major point of this study. Thus, further studies are necessary to demonstrate the significance of this work. I cannot support publication at the current stage and the manuscript could be further considered if the authors could address the following concerns:

1. Line 64-66, 'RT synthesis in open air is not feasible for the synthesis of pure-iodine perovskite CsPbI₃ QDs'. This is inaccurate. As mentioned above, uniform cubic-phase CsPbI₃ NCs with yellow emission synthesized at room temperature in open air has been reported in Nanoscale, 2021, 13, 4899-4910.
2. Line 81-83, '...the studies on direct and rapid RT alloying of CsPbI₃ or CsSnI₃ QDs in open air have never been reported before. Moreover, it is difficult to prepare stable sub-5 nm sized iodine-based perovskite QDs...' This is inaccurate. The idea of room temperature synthesis of Sn containing nanocrystals has been widely studied, including direct synthesis (Nat. Photon. 2021, 15, 696-702, Chem. Mater. 2020, 32, 1089-1100, Light Sci. Appl. 2023, 12, 208) and post-synthesis Sn-Pb alloying (ACS Energy Lett. 2017, 2, 1190-1196, J. Am. Chem. Soc. 2017, 139, 11, 4087-4097). Pure iodine-based perovskite quantum dots of less than 3 nm have also been successfully synthesized (ACS Energy Lett. 2017, Light Sci. Appl. 2023, 12, 208). The author also mentioned that their RT synthesis strategy is also applicable to pure-iodine hybrid organic-inorganic perovskite QDs (Line 98-99). Please compare the synthesis methods in detail for Sn containing perovskite nanocrystals and demonstrate why their method is unique for achieving higher stability.
3. Figure S3 shows the yellow emission from FA-based Sn-Pb iodide hybrid perovskite QDs. The authors need to give the TEM image with size distribution to show the morphology of the nanocrystals for the hybrid system. Are they still spherical quantum dots? In addition, please give the accurate Sn/Pb elemental ratio of the precursor and the product.
4. Line 192-194, 'the PL emission intensity of Sn-Pb perovskite QDs is much higher than that of CsPbI₃ QDs, as can be seen from the photographs of the colloidal solutions under UV light'. The authors used photographs to compare the PL emission of Pb and Sn-Pb QDs, which is a very irregular way. I would suggest the authors to measure the absolute photoluminescence quantum yield (PLQY) of the system to compare the emission efficiencies. Many reports are showing near-unity PLQY of CsPbI₃ NCs (ACS Nano 2017, 11, 10, 10373-10383, J. Chem. Phys. 2020, 152, 020902) whereas Sn-Pb nanocrystals usually have a PLQY less than 0.3% (Angew. Chem. 2020, 132, 8499-8502). Could the authors comment on how their synthesis overcomes the low quantum yield in Sn-Pb perovskite nanocrystals?

5. Line 211-215, The author mentioned that defect passivation through stannous alloying can passivate the uncoordinated Pb atoms in the crystal lattice. Is this correct? I do not see evidence that stannous can coordinate with the uncoordinated Pb atoms. Could the authors give a diagram of crystal structure with the incorporation process and describe how Sn²⁺ can coordinate with Pb²⁺?
6. Figure S10 and S11 show the absorption and photoluminescence of nanocrystals synthesized at higher temperatures. Clearly, the products are a mixture of 3 to 4 phases/nanostructures. The authors only comment on the primary peak, which is at around 600 nm, but the other peaks are still very sharp (for example, the peaks at 560 nm, 620 nm, 650 nm for Figure S10b, the peaks at 560 nm, 620 nm for Figure S10e, Figure S11b, and the peaks at 620 nm for Figure S11e). What are these peaks? Do these peaks result from Sn - Pb phase segregation or Sn²⁺ - Sn⁴⁺ oxidation? If a mixture system with unknown composition is photoexcited, then the PL lifetime is meaningless. The authors need to determine the exact composition for each phase and get reliable results from a clean system.
7. Fig. 3b shows that the pure-iodine Sn-Pb perovskite QDs possess a direct bandgap of 2.12 eV. What is the composition (Sn/Pb ratio) of the system used for DFT calculation here? Is the Sn/Pb ratio for DFT consistent with the experimental composition? How do the authors consider the Sn²⁺ to Sn⁴⁺ oxidation in the DFT, which is an unavoidable but critical factor for Sn²⁺ perovskites?
8. In addition to the previous question, I do not see any results characterizing the Sn/Pb ratio in the article, and the entire article does not give the chemical formula of the Sn-Pb material system studied. The authors should give a formula (or Sn/Pb ratio) of the perovskite nanocrystals. Probably extremely low doping of Sn is the reason for the slight change in bandgap: Line 221-224, 'no noticeable change in the exciton absorption and PL emission characteristics of the as-prepared Sn-Pb perovskite QDs with the increase of the Pb/Sn ratio.'
9. Line 291-293, 'QDs with longer emission wavelengths lose their PL performance after being placed in air for one day, and their RT stability is much lower than that of wider bandgap QDs. Therefore, the colloidal stability of the pure-iodine perovskite QDs largely benefits from the small QD size.' The authors missed the information on what the stability is for wider bandgap QDs; please attach a stability test of both systems. The authors mentioned that smaller QDs are more colloidal stable, but in Line 283-285 they mentioned that small QDs have much more traps than larger ones. The authors need to explain why small QDs with more traps (more Cs, Pb or halide vacancies in the structure) show much better stability.
10. The authors used the model of passivation of Pb vacancy defect to explain the increased stability of Sn-Pb perovskites in Line 328-335. This trend is exactly opposite to the trend of stability and defect states mentioned by the author in the previous question, where larger NCs with less defect show much worse stability. In addition, the Sn²⁺ can easily change into Sn⁴⁺ even in a glovebox with several ppm of O₂. The authors need to consider this practical intrinsic instability of Sn either in the simulation or in the discussions. How will this affect the stability of Sn-Pb perovskites?
11. Line 335-336, 'the formation of vacancy defects gradually becomes difficult with the increased proportion of stannous ions in the Sn-Pb iodide perovskite'. The authors claimed that the increased proportion of Sn can stabilize the structure (better stability than Pb) by decreasing vacancy defects. However, in Line 226-229, the authors mentioned that excessive Sn leads to more defect states. ('excessive incorporation of stannous ions will cause an increase in defect states of Sn-Pb perovskite QDs, leading to intensified nonradiative recombination and PL quenching.'). This seems to be quite contradictory. A detailed relationship between stability, Sn/Pb ratio, vacancy traps, and Sn⁴⁺ content should be given to support the experimental results.
12. Regarding the oxidation from Sn²⁺ to Sn⁴⁺, the author only mentioned in Line 169-171 that 'Unfortunately, the purified pure-tin perovskite QD colloidal solution cannot be obtained in open air due to the susceptibility of stannous ions to oxygen gas.'. Sn⁴⁺ plays a critical role in these nanocrystals synthesized in open air. The authors need to give Pb/Sn ratio and the Sn²⁺/Sn⁴⁺ ratio from XPS measurements (Figure S8) and ensure they are focusing on incorporating Sn²⁺ as they claimed.

There are some minor issues:

1. There is another small bump for both Br (at 500 nm) and Cl (at 410 nm) samples. The authors need to make sure Br and Cl samples do not have phase segregation. In addition, the

- photoluminescence of Br and Cl based Sn-Pb nanocrystals should also be given in Figure S4.
2. In Figure S8b, the caption (Br 3d) does not match the label in the map (I 3d), please correct this.
 3. All the 'Sn-Pb QDs' should be replaced by the exact chemical formula of the perovskites.

Point-by-point responses to the reviewers

We appreciate the suggestions and the comments raised by the reviewers. We have carefully addressed these comments and responded accordingly. We have incorporated all the suggestions and addressed all comments in our revised manuscript to make our conclusions more convincing. The main corrections in the revised manuscript are marked in red, and a comprehensive point-by-point response to the reviewers is given below.

Response to Reviewer #1 :

In this manuscript, Zhang et al present a room-temperature synthesis approach for strongly confined colloidal Cs(Pb:Sn)I₃ nanocrystals (NCs). The main driving parameter for the quantum confinement seems to be the Cs/Pb precursors' ratio. While the discussed results are of interest, I have several minor and major comments that, in my opinion, should be addressed before considering publication.

Response: *We sincerely thank the reviewer for the thorough and careful reading of our manuscript and appreciate the reviewer for the positive comments and valuable insights.*

1. Logical order: The tuning of the NC size by controlling Cs/Pb feed ratio is only presented in Figure 3, while Figures 1 and 2 deal already with an “optimized” ratio (1:4) whose origin the reader cannot understand at this stage. I would recommend first showing and discussing the effect of Cs/Pb ratio to get sub-5nm NCs and then focusing on the chosen ratio of 1:4. In a sense this means starting with Figure 3.

Response: *We thank the reviewer for the suggestion. In this work, our research core was to adopt a completely new reaction system to prepare strongly confined all-inorganic pure-iodine tin-lead (Sn-Pb) alloyed perovskite quantum dots (QDs) at room temperature in open air. Fig.1 showed the*

schematic illustration and reaction mechanism of the direct room temperature strongly confined colloidal synthesis, and the basic characteristics (TEM image and optical spectra) of the typical pure-iodine all-inorganic Sn-Pb perovskite QDs with a wide bandgap and sub-5 nm size. Then Fig. 2 further indicates that the luminescence performance of this yellow emissive Sn-Pb perovskite QDs is significantly better than that of the pristine CsPbI₃ QDs, confirming the key role of stannous ions alloyed in perovskite crystals in improving PL performance. In order to further reveal the origin of this strong confinement effect, the Cs/Pb ratio in the precursors was regulated and the optical experiment results were shown in Fig.3. So in our opinion, the contents in Figs. 1 and 2 placed at the front could fully demonstrate the novelty (strongly confined synthesis and direct room temperature Sn/Pb alloying for size-stabilized pure-iodine all-inorganic Sn-Pb mixed perovskite QDs) and the characteristics of this work, Fig. 3 is the supplementary evidence for Figs. 1 and 2, aiming to reveal the key factor affecting the quantum-confined effect of all-inorganic pure-iodine Sn-Pb perovskite colloidal QDs. Finally, we still maintain the original graph order, and more relevant explanations have been provided in the revised version. Much thanks to the reviewer for this suggestion.

2. Figure 3c shows clear changes in PL. Ratios of 1:1 and 1:2 show broad and multi-peak signals pointing towards high size polydispersity. TEM should be included to evaluate the NC sizes versus precursors' ratios. Also, given the abrupt shift from 1:2 to 1:4, it may be worth exploring an intermediate ratio of 1:3.

Response: *We thank the reviewer for the insightful comments. TEM images of the samples with different Cs/Pb ratios have been provided in Supporting Information. As expected by the reviewer, the NC sizes of the resultant samples with Cs/Pb ratios of 1:1 and 1:2 show high polydispersity, as*

displayed in Fig. R1b and c. The samples with Cs/Pb ratios of 1:1 and 1:2 have different morphologies, such as nanoparticles, nanosheets, and nanorods. The multimodal nature of the PL spectra implied the presence of NCs with different morphologies and sizes in the reaction products. TEM image of the sample with an intermediate ratio of 1:3 is also displayed in Fig. R1a. It is found that the samples with ratios of 1:4 and 1:3 exhibit similar morphology and size.

Fig. R1 TEM images of pure-iodine all-inorganic Sn-Pb perovskite QDs with varying Cs/Pb feed molar ratio.

3. The manuscript discusses results as a function of elemental feed ratios but there is no quantitative or semiquantitative analysis of the final perovskite composition. I believe this is very important to analyze the results. Keep in mind that bandgap energy of Pb-Sn perovskites is not a monotonous function of Pb:Sn ratio (bandgap bowing). Actually, the authors present XPS results, from which semi-quantitative analysis should be easy to extract.

Response: We thank the reviewer for the insightful comments. The semi-quantitative results for the percentage of Sn/(Sn+Pb) from XPS measurement was about 7.2%. To further accurately reveal the

composition of the alloyed perovskite QDs, the quantitative analysis of all-inorganic pure-iodine Sn-Pb perovskite QDs was performed by inductively coupled plasma-optical emission spectrometry (ICP-OES). The ICP-OES measurement result shows that the proportion of Sn/(Sn+Pb) in yellow emissive sample synthesized using this room temperature strongly confined strategy was 9.1%. Therefore, the chemical formula of pure-iodine Sn-Pb perovskite QDs in the manuscript is represented as $CsSn_{0.09}Pb_{0.91}I_3$.

4. In Figure S8, caption for panel (b) should be I 3d instead of Br 3d.

Response: We thank the reviewer for pointing out it. The caption of panel (b) in Figure S8 has been corrected.

Response to Reviewer #2:

The manuscript reported a RT strongly confined spontaneous crystallization strategy in open air for preparing sub-5 nm sized pure-iodine tin-lead (Sn-Pb) perovskite colloidal QDs with bright yellow luminescence. However, as an important part of this manuscript, this synthesis method lacks detailed description and analysis of its process. And the novelty of this work is not significant. Moreover, the analysis of some test data is insufficient. Therefore, I would recommend to reject this manuscript considering that Nature Communications is targeting high impact journal with broad readership. I think the following issues should be addressed seriously, before the next submission.

Response: We sincerely thank the reviewer for the thorough and careful reading of our manuscript and appreciate the reviewer for the suggestions. The suggestions are indeed beneficial for improving the quality of our manuscript. Firstly, we have added detailed description and analysis for the synthesis method. Secondly, to strengthen the novelty of this work, we added more

explanation and analysis in the revised edition.

All-inorganic halide perovskite QDs/NCs exhibit considerable potential applications in optoelectronics. However, compared to the conventional hot-injection method, room temperature synthesis of photoactive pure-iodine all-inorganic perovskite QDs/NCs is still a challenge due to the thermal unequilibrium-induced metastable (perovskite)-to-stable (non-perovskite) phase transition. Besides, room temperature synthesis without inert gas protection has attracted more attention due to its easy manipulation and mild reaction conditions. Thus, it is very necessary to develop a simple and feasible room temperature synthesis approach for synthesizing all-inorganic pure-iodine perovskite NCs.

Even more significant, the studies on direct and rapid room temperature alloying of CsPbI₃ or CsSnI₃ QDs have never been reported before. As a result, the novelty of our work can be summarized as follows: 1) A completely new room temperature reaction strategy with simple preparation of precursor solutions was proposed for pure-iodine all-inorganic alloyed perovskite colloidal QDs. The entire synthesis processes could be carried out in a room temperature atmospheric environment. 2) Because the crystallization rate of halide perovskite QDs is extremely fast, it is quite easy for iodine-based reaction system to directly form a stable nonphotoactive yellow phase with a larger crystal size through a room temperature process. However, the room temperature stability of perovskite-phase iodine-based QDs can be highly improved by the reduced QD size, strong quantum confinement effect and increased surface Gibbs energy. In this work, the yellow emissive pure-iodine all-inorganic perovskite QDs were successfully synthesized through room temperature strongly confined spontaneous crystallization in open air. The successful preparation of photoactive perovskite-phase iodine-based all-inorganic QDs benefited from strong size confinement and enhanced phase stability. 3) Based on this size-stabilized synthesis

process, the pure-iodine all-inorganic Sn-Pb perovskite QDs were directly prepared for the first time at room temperature. The resultant pure-iodine all-inorganic Sn-Pb alloyed perovskite QDs exhibited the enhanced PL intensity, prolonged fluorescence lifetime and improved colloidal stability as compared with CsPbI₃ QDs. This work opens an effective way for the direct room temperature alloying of pure-iodine all-inorganic CsPbI₃ QDs with CsSnI₃, enriches the synthesis pathways of pure-iodine all-inorganic Sn-Pb mixed perovskite QDs, and fills the gap in related research. The results in our work have not been reported in literature. Based on these reasons, we strongly believe that our work has sufficient novelty and presents substantial breakthrough in this field.

1. Detailed description and analysis of the process of the proposed “RT strongly confined spontaneous crystallization strategy” should be provided. This strategy is the key part of this paper, and more space should be devoted to discussing the reaction mechanism or some phenomena in the reaction process.

Response: *We thank the reviewer for the insightful comments. The detailed description and analysis of “RT strongly confined spontaneous crystallization strategy” have been presented in Fig. 1a, Experimental Section in Supporting Information and the revised manuscript. Thanks to the extremely fast crystallization rate of halide perovskite QDs, it is very easy for iodine-based reaction system to directly form a stable nonphotoactive yellow phase with a large crystal size through a room temperature process. However, colloidal CsPbI₃ QDs with small crystal size have shown a much more stable perovskite structure due to the strong quantum confinement effect and high surface Gibbs energy from the dominant contribution of large surface/volume ratio (Adv. Mater. 2020, 32, 2002632; J. Mater. Chem. A, 2020, 8, 10226-10232; Science 2016, 354, 92-95). Therefore,*

in order to obtain a sufficiently stable Sn²⁺ alloyed CsPbI₃ perovskite phase structure at room temperature, reducing the crystal size through room temperature strongly confined strategy is crucial. Relatively speaking, larger CsPbI₃ perovskite crystals are more likely to undergo phase transitions, resulting in the disappearance of photoactivity. Therefore, the crystal size of pure-iodine all-inorganic perovskite QDs plays a dominant role in room temperature stability.

In this work, the feed ratio of Cs/Pb precursors plays a crucial role in size confinement and PL emission wavelength of the resultant QDs. In general, when the amount of a reactant (Cs⁺ precursor) is relatively low, the reaction rate of the entire reaction system will slow down, thereby severely inhibiting the growth rate of QD crystals. In other words, the resultant perovskite colloidal QDs shrink in size with a decreased Cs/Pb feed molar ratio, and exhibit strong size confinement. As a result, the pure-iodine all-inorganic perovskite QDs produced in a Cs-deficient reaction system show superior phase stability of perovskite structure. Based on this size-stabilized synthesis process, the pure-iodine all-inorganic Sn-Pb perovskite QDs are successfully prepared at room temperature for the first time.

Our RT synthesis process was very simple and rapid. As shown in Fig. 1a, the Cs, Pb, and Sn precursors were easily soluble in oleic acid (OA)/toluene mixed solution, when trimethylsilyl iodide (TMSI) solution was added into the mixed solution containing precursor-ligand complexes (Cs/Pb precursor ratio of 1:4), a blood-red solution appeared immediately and emitted bright yellow luminescence under 365 nm ultraviolet (UV) excitation, as shown in Fig. S1. Due to the fact that metal halide perovskite is an ionic crystal with very low solubility in non-polar solvents, the constituent elements of perovskite QDs undergo spontaneous crystal growth in non-polar toluene solvent. However, the stable non-perovskite δ -phase without photoactive properties would be directly formed with a higher Cs/Pb feed molar ratio, as shown in Fig. R6. The colors under

daylight of the crude solutions for these two products are completely different. The possible reaction mechanism of QD growth is illustrated on the right side of Fig. 1a. The iodine ions in TMSI were released through the formation of alkyl ammonium halides, which initiated a rapid salt metathesis reaction and resulted in the nucleation and growth of pure-iodine all-inorganic Sn-Pb alloyed perovskite QDs.

2. The authors claimed that the prepared pure-iodine tin-lead (Sn-Pb) perovskite colloidal QDs show highly bright yellow luminescence. The most fundamental and direct data of photoluminescence quantum yield should be provided to support this claim.

Response: We thank the reviewer for the suggestion. The photoluminescence quantum yield (PLQY) measurement was performed for the resultant QDs. The PLQY values of CsPbI₃ QDs and pure-iodine all-inorganic Sn-Pb perovskite QDs are 21.7% and 55.4%, respectively. It also indicated that the emission efficiency of all-inorganic pure-iodine Sn-Pb alloyed perovskite QDs is significantly higher than that of the pristine CsPbI₃ QDs.

3. In Figure S9b, the PL spectra exhibit an obvious blueshift with the increase of the Pb/Sn ratio, which is inconsistent with the author's description that PL emission properties do not change.

Response: We thank the reviewer for the suggestion. A deviation of several nanometers in the emission peak position indeed exists with the increase of the Pb/Sn ratio. However, their absorption spectra and excited fluorescence are almost identical. The minor shift for PL emission peaks may come from the slight variation in the incorporation amount of stannous ions. Therefore, the relevant description has been corrected to “the exciton absorption and PL emission characteristics of the as-prepared Sn-Pb perovskite QDs show negligible changes with the increase of the Pb/Sn ratio”.

4. The proof for the formation of Sn-Pb alloyed perovskite QDs is insufficient. More characterizations, such as element mapping and lattice spacing, should be provided.

Response: We thank the reviewer for the insightful comments. The element mapping and lattice spacing are provided for pure-iodine all-inorganic Sn-Pb perovskite QDs, as shown in Fig. R2 and Fig. R3, which validate the formation of pure-iodine all-inorganic Sn-Pb perovskite QDs. Besides, ICP-OES and XPS measurements have further confirmed the successful preparation of pure-iodine all-inorganic Sn-Pb perovskite QDs by this RT spontaneous crystallization strategy.

Fig. R2 Elemental mapping images of an individual element in the pure-iodine all-inorganic Sn-Pb perovskite QDs with Cs/Pb ratio of 3:2.

Fig. R3 HRTEM images of (a) CsPbI₃ QDs and (b) pure-iodine all-inorganic Sn-Pb perovskite QDs, indicating the lattice contraction of the perovskite structure with the incorporation of stannous ions.

5. In Figure 3c, the PL spectra are broadened and have a tendency of multiple peaks when the Cs/Pb ratio is 1:1 and 1:2. Reasonable explanations should be added.

Response: We thank the reviewer for the insightful comments. Reasonable explanations have been

added in this revised edition. As expected by **Reviewer #1**, the NC sizes of the resultant samples with Cs/Pb ratios of 1:1 and 1:2 show high polydispersity, as displayed in Fig. R1b and c. The samples with Cs/Pb ratios of 1:1 and 1:2 have different morphologies, such as nanoparticles, nanosheets, and nanorods. The multimodal nature of the PL spectra implied the presence of NCs with different morphologies and sizes in the reaction products and size-dependent effect. The size confinement effect on the optical bandgap can be accounted for in a spherical potential well according to $\Delta E_{\text{gap}} = \hbar^2 \pi^2 / (2\mu r^2)$, where r is the particle radius and μ is the reduced mass of the exciton. This provides favorable evidences for size-dependent effect and PL multimodal nature of semiconductor QDs. In fact, the broad and multi-peak signals were widely present in the studies of lead halide perovskite NCs, as described in the following reported papers (*J. Phys. Chem. Lett.* 2019, 10, 4149-4156; *Chem. Mater.* 2019, 31, 365-375; *Nano Lett.* 2018, 18, 1246-1252; *Adv. Mater.* 2022, 34, 2107105; *Adv. Funct. Mater.* 2022, 32, 2108687).

Response to Reviewer #3:

In this manuscript, the authors reported yellow emission from pure-iodine all-inorganic perovskite QDs/NCs synthesized at room temperature and observed that stannous ions can passivate the lead vacancies, leading to enhanced stability. The key point in this manuscript is the room-temperature synthesis of sub-5 nm sized pure-iodine QDs that showed peak PL at 563 nm, just as the authors claimed in the manuscript - 'yellow emission from pure-iodine QDs/NCs is experimentally achieved for the first time'. First of all, the results in this manuscript are very similar to the reported work (*Nanoscale*, 2021, 13, 4899-4910). The *Nanoscale* paper reported uniform cubic-phase CsPbI₃ NCs with PL peak at 569 nm, which are also synthesized at room temperature in open air (i.e. same nanocrystals, same synthesis condition, and very similar yellowish emission). In addition, the

Nanoscale paper also demonstrated the ultrasmall-sized CsPbBr₃ NCs with emission at 470 nm and ultrasmall-sized CsPbCl₃ NCs with emission at 385 nm, where uniform nucleation and growth and chemical stability are simultaneously achieved with their method. In addition to this, I would also like to make the authors aware of the Sn-Pb perovskite nanocrystal papers where room temperature synthesis has been developed and ultrasmall Sn-Pb dots of <3 nm has been demonstrated (ACS Energy Lett. 2017, 2, 1190-1196, Nat. Photon. 2021, 15, 696-702, Chem. Mater. 2020, 32, 1089-1100, Light Sci. Appl. 2023, 12, 208). The authors should make clear the novelty of their study, by careful comparison with the synthesis method in the papers listed above, as this idea is the major point of this study. Thus, further studies are necessary to demonstrate the significance of this work. I cannot support publication at the current stage and the manuscript could be further considered if the authors could address the following concerns.

Response: We sincerely thank the reviewer for the thorough and careful reading of our manuscript and appreciate the reviewer for the suggestions. To strengthen the novelty of this work, we added more explanation and analysis in this revised edition. All-inorganic Cs-based halide perovskite QDs/NCs exhibit considerable potential applications in optoelectronics. However, compared to the conventional hot-injection method, room temperature synthesis of photoactive pure-iodine all-inorganic perovskite QDs/NCs is still a big challenge due to the thermal unequilibrium-induced metastable (perovskite)-to-stable (non-perovskite) phase transition. Besides, room temperature synthesis without inert gas protection has attracted more attention due to its easy manipulation and mild reaction conditions. Thus, it is very necessary to develop a simple and feasible room temperature synthesis approach for synthesizing all-inorganic Cs-based pure-iodine perovskite NCs. Even more significant, the studies on direct and rapid room temperature alloying of all-inorganic CsPbI₃ or CsSnI₃ QDs have never been reported before. As a result, the novelty of

our work can be summarized as follows: 1) A **completely new room temperature reaction system with simple preparation of precursor solutions** was proposed for pure-iodine all-inorganic perovskite colloidal QDs. The entire synthesis processes could be carried out in a room temperature atmospheric environment. 2) Because the crystallization rate of halide perovskite QDs is extremely fast, it is very easy for iodine-based reaction system to directly form a stable nonphotoactive yellow phase with a larger crystal size through a room temperature process. However, **the room temperature stability of perovskite-phase iodine-based QDs can be highly improved by the reduced QD size, strong quantum confinement effect and increased surface Gibbs energy**. In this work, **the yellow emissive pure-iodine all-inorganic perovskite QDs were successfully synthesized through room temperature strongly confined spontaneous crystallization in open air**. The successful preparation of photoactive perovskite-phase iodine-based all-inorganic QDs benefited from strong size confinement and enhanced phase stability. 3) Based on this size-stabilized synthesis process, **the pure-iodine all-inorganic Sn-Pb perovskite QDs were directly prepared for the first time at room temperature**. The resultant all-inorganic Sn-Pb perovskite QDs exhibited the enhanced PL intensity, prolonged fluorescence lifetime and improved colloidal stability as compared with CsPbI₃ QDs. This work opens an effective way for the direct room temperature alloying of pure-iodine Cs-based Sn-Pb mixed perovskite QDs, enriches the synthesis pathways of pure-iodine Cs-based Sn-Pb perovskite QDs, and fills the gap in related research.

We have carefully read the related papers mentioned by the reviewer, a detailed comparison was made on these synthesis processes and composition of pure-iodine perovskite QDs, as listed in Table R1. It should be pointed out that **our key point of this work is not how small the size of pure-iodine QDs is, but rather a completely new room temperature reaction system for high-quality pure-iodine Cs-based Sn-Pb perovskite QDs, with the assistance of size-stabilized**

perovskite phase and Sn/Pb direct alloying. This work has pioneered the direct room temperature preparation of pure-iodine all-inorganic Sn-Pb alloyed perovskite QDs. However, the literature mentioned by the reviewer mostly involves the hot-injection synthesis process and organic-inorganic hybrid perovskite NCs (ACS Energy Lett. 2017, 2, 1190-1196, Nat. Photon. 2021, 15, 696-702, Chem. Mater. 2020, 32, 1089-1100, Light Sci. Appl. 2023, 12, 208), which is not suitable for comparison with our room temperature preparation method for pure-iodine Cs-based perovskite QDs. Even more remarkably, room temperature synthesis of pure-iodine Cs-based perovskite QDs is more difficult compared to organic-inorganic hybrid counterparts, because the smaller ionic size of Cs⁺ is not enough to maintain the lead-iodine octahedron to form a periodic perovskite structure.

Table R1. Comparisons of synthesis technology for pure-iodine perovskite QDs in the mentioned references and our study.

Perovskite	Synthetic method	Reactants	PL wavelength (nm)	Ref.	Remarks
CsPbI ₃	Room temperature synthesis	Cs ₂ CO ₃ , PbO, triphenylphosphine (Ph ₃ P), oleic acid (OA), oleylamine (OAm), stearic acid, hexadecylamine, mesitylene, solid iodine (I ₂), chloroform (CH ₃ Cl), dichloromethane (CH ₂ Cl ₂), toluene	~570 - 620 (orange-red emission)	Nanoscale, 2021, 13, 4899-4910	Need to use Schlenk-line equipment to prepare metal precursor (relatively complex preparation processes of precursor solutions)
CH ₃ NH ₃ SnI ₃ CH ₃ NH ₃ Sn _x Pb _{1-x} I ₃	Hot-injection route + Cation exchange (70 °C)	SnI ₂ , trioctylphosphine (TOP), octadecene, OA, OAm, methylamine, tetrahydrofuran, PbI ₂	~730 - ~760	ACS Energy Lett. 2017, 2, 1190-1196	Not a room temperature method, and not all-inorganic perovskite QDs
CH(NH ₂) ₂ SnI ₃	Hot-injection route with varying reaction temperature of the precursors	SnI ₂ , TOP, formamidinium acetate, octadecene, toluene, OA, OAm	~660 - ~770 (Size: 7.3±0.8 nm to 12.1±1.1 nm)	Nat. Photon. 2021, 15, 696-702	Including 25 °C room temperature synthesis under an inert atmosphere, but not all-inorganic Pb-based perovskite QDs
CsPbI ₃ CsSn _x Pb _{1-x} I ₃	Hot-injection route	PbI ₂ , trioctylphosphine oxide (TOPO), Cs ₂ CO ₃ , OA, OAm, octadecene, SnI ₂	~690	Chem. Mater. 2020, 32, 1089-1100	Not a room temperature method

$CsSn_xPb_{1-x}I_3$	Hot-injection route	Cs_2CO_3 , OA, OAm, octadecene, PbI_2 , SnI_2 , TOP	/	Light Sci. Appl. 2023, 12, 208	Not a room temperature method
$CsSn_{0.09}Pb_{0.91}I_3$	Room temperature synthesis	Cesium acetate, lead acetate, tin(II) 2-ethylhexanoate, toluene, OA, OAm, iodotrimethylsilane	563	This work	An efficient and facile room temperature method, all-inorganic Sn-Pb perovskite QDs

Although the room temperature synthesis of $CsPbI_3$ NCs with orange-red emission was demonstrated in *Nanoscale* paper, the preparation of metal precursor requires a Schlenk-line equipment with heating and inert atmosphere, which is relatively complex and inconvenient as compared with our preparation process. Besides, the yellow emission in our work is significantly different from the orange-red fluorescence (see the digital fluorescence photographs of Fig. 1 in *Nanoscale* paper). More importantly, the main point of *Nanoscale* paper is to study the influence law of the cationic ligands or reaction solvent on the $CsPbX_3$ products, and the direct room temperature preparation of pure-iodine Cs-based Sn-Pb alloyed perovskite QDs is not involved. However, the main task of this work is to develop a superior room temperature synthesis pathway for pure-iodine Sn-Pb alloyed perovskite QDs. Therefore, we believe that our work is very different from the mentioned references and the related results in our work have been rarely reported in literature.

Based on these reasons, we strongly believe that our work has sufficient novelty and presents substantial breakthrough, which meets the high standards of *Nature Communications*.

1. Line 64-66, 'RT synthesis in open air is not feasible for the synthesis of pure-iodine perovskite $CsPbI_3$ QDs'. This is inaccurate. As mentioned above, uniform cubic-phase $CsPbI_3$ NCs with yellow emission synthesized at room temperature in open air has been reported in *Nanoscale*, 2021, 13, 4899-4910.

Response: We thank the reviewer for the suggestion. We agree with the reviewer's viewpoint. This

sentence has been corrected as “RT synthesis in open air is an enormous challenge for pure-iodine perovskite CsPbI₃ QDs”.

2. Line 81-83, ‘...the studies on direct and rapid RT alloying of CsPbI₃ or CsSnI₃ QDs in open air have never been reported before. Moreover, it is difficult to prepare stable sub-5 nm sized iodine-based perovskite QDs...’ This is inaccurate. The idea of room temperature synthesis of Sn containing nanocrystals has been widely studied, including direct synthesis (Nat. Photon. 2021, 15, 696-702, Chem. Mater. 2020, 32, 1089-1100, Light Sci. Appl. 2023, 12, 208) and post-synthesis Sn-Pb alloying (ACS Energy Lett. 2017, 2, 1190–1196, J. Am. Chem. Soc. 2017, 139, 11, 4087–4097). Pure iodine-based perovskite quantum dots of less than 3 nm have also been successfully synthesized (ACS Energy Lett. 2017, Light Sci. Appl. 2023, 12, 208). The author also mentioned that their RT synthesis strategy is also applicable to pure-iodine hybrid organic-inorganic perovskite QDs (Line 98-99). Please compare the synthesis methods in detail for Sn containing perovskite nanocrystals and demonstrate why their method is unique for achieving higher stability.

Response: *We thank the reviewer for the suggestion. Firstly, it should be noted that the research content of this work mainly focuses on the all-inorganic Cs-based iodide perovskite QDs and their direct preparation of room temperature alloying. In addition, the ionic size of Cs⁺ is not as large as that of CH(NH₂)₂⁺ (FA⁺) or CH₃NH₃⁺ (MA⁺) organic cations, which is not enough to maintain the lead-iodine octahedron to form a periodic perovskite structure, resulting in the undesirable crystal distortion and structural transformation. Thus, the room temperature synthesis of all-inorganic Cs-based perovskite QDs is more difficult than that of organic-inorganic hybrid perovskite QDs. As shown in Table R2, the preparation methods in those papers mentioned by the reviewer are not the direct room temperature synthesis process, but all involve the hot-injection synthesis process. In*

addition, some of the papers (*Nat. Photon.* 2021, 15, 696-702, *ACS Energy Lett.* 2017, 2, 1190–1196) focus on organic-inorganic hybrid perovskite rather than the all-inorganic Cs-based iodide perovskite we have studied. There are significant differences in types of perovskite and preparation processes. Therefore, it does not make much sense to compare them. Besides, it also indicates that the studies on **direct and rapid room temperature alloying** of all-inorganic CsPbI₃ with CsSnI₃ QDs in open air have never been reported before.

Table R2. Comparisons of synthesis technology for pure-iodine Sn-containing perovskite QDs in the mentioned references and our study.

Perovskite	Synthetic method	Reactants	Ref.	Remarks
$CH(NH_2)_2SnI_3$	Hot-injection route with varying reaction temperature of the precursors	SnI ₂ , TOP, formamidinium acetate, octadecene, toluene, OA, OAm	Nat. Photon. 2021, 15, 696-702	Including 25 °C room temperature synthesis under an inert atmosphere, but not all-inorganic Pb-based perovskite QDs
$CsSn_xPb_{1-x}I_3$	Hot-injection route	PbI ₂ , trioctylphosphine oxide (TOPO), Cs ₂ CO ₃ , OA, OAm, octadecene, SnI ₂	Chem. Mater. 2020, 32, 1089-1100	Not a room temperature method
$CsSn_xPb_{1-x}I_3$	Hot-injection route	Cs ₂ CO ₃ , OA, OAm, octadecene, PbI ₂ , SnI ₂ , TOP	Light Sci. Appl. 2023, 12, 208	Not a room temperature method
$CH_3NH_3Sn_xPb_{1-x}I_3$	Hot-injection route + Cation exchange (70 °C)	SnI ₂ , trioctylphosphine (TOP), octadecene, OA, OAm, methylamine, tetrahydrofuran, PbI ₂	ACS Energy Lett. 2017, 2, 1190-1196	Not a direct room temperature method, and not all-inorganic perovskite QDs
$CsPb_{1-x}Sn_xBr_3$	Hot-injection route + Cation exchange (stirred at room temperature for ~16 h)	Cs ₂ CO ₃ , OA, OAm, octadecene, PbBr ₂ , SnBr ₂ , toluene	J. Am. Chem. Soc. 2017, 139, 11, 4087–4097	Not a direct room temperature method, and not iodine-based perovskite QDs
$CsSn_{0.09}Pb_{0.91}I_3$	Direct room temperature synthesis	Cesium acetate, lead acetate, tin(II) 2-ethylhexanoate, toluene, OA, OAm, iodotrimethylsilane	This work	An efficient and facile room temperature method, all-inorganic Sn-Pb perovskite QDs

Thanks to the extremely fast crystallization rate of halide perovskite QDs, it is very easy for iodine-based reaction system to directly form a stable nonphotoactive yellow phase with a larger

crystal size through a room temperature process. However, the shortage in stability of perovskite-phase iodine-based QDs can be highly improved by room temperature strongly confined spontaneous crystallization strategy. Based on the size-stabilized synthesis process, the pure-iodine all-inorganic perovskite QDs show significantly increased colloidal stability due to the passivation effect of stannous ions and enhanced formation energy, resulting from the direct Sn/Pb alloying.

3. Figure S3 shows the yellow emission from FA-based Sn-Pb iodide hybrid perovskite QDs. The authors need to give the TEM image with size distribution to show the morphology of the nanocrystals for the hybrid system. Are they still spherical quantum dots? In addition, please give the accurate Sn/Pb elemental ratio of the precursor and the product.

Response: We thank the reviewer for the insightful comments. The TEM image of FA-based Sn-Pb iodide hybrid perovskite QDs with size distribution has been added, as shown in Fig. R4. The morphology of hybrid perovskite QDs is similar to that of all-inorganic Cs-based counterparts, and the QDs still exhibit spherical in shape with an average size of 3.59 nm. The Sn/Pb elemental ratio in the precursors is 3:4, and the actual proportion of Sn/(Sn+Pb) in the product is 10.8%, which is determined by ICP-OES. Finally, the corresponding modifications have been made in the Supporting Information.

Fig. R4 TEM image and the corresponding HRTEM image and size distribution of FA-based Sn-Pb iodide hybrid perovskite QDs.

4. Line 192-194, 'the PL emission intensity of Sn-Pb perovskite QDs is much higher than that of CsPbI₃ QDs, as can be seen from the photographs of the colloidal solutions under UV light'. The authors used photographs to compare the PL emission of Pb and Sn-Pb QDs, which is a very irregular way. I would suggest the authors to measure the absolute photoluminescence quantum yield (PLQY) of the system to compare the emission efficiencies. Many reports are showing near-unity PLQY of CsPbI₃ NCs (ACS Nano 2017, 11, 10, 10373-10383, J. Chem. Phys. 2020, 152, 020902) whereas Sn-Pb nanocrystals usually have a PLQY less than 0.3% (Angew. Chem. 2020, 132, 8499-8502). Could the authors comment on how their synthesis overcomes the low quantum yield in Sn-Pb perovskite nanocrystals?

Response: *We thank the reviewer for the suggestion. Firstly, it should be noted that PL spectra of CsPbI₃ QDs and pure-iodine all-inorganic Sn-Pb perovskite QDs can intuitively reflect the intensity of their PL emission. Fig. 2c shows that the PL emission intensity of Sn-Pb perovskite QDs is much higher than that of CsPbI₃ QDs, and the photographs of the colloidal solutions under UV light further support the evidence above. In addition, the absolute PLQY measurement was further carried out. The PLQY values of CsPbI₃ QDs and pure-iodine all-inorganic Sn-Pb perovskite QDs are 21.7% and 55.4%, respectively. These results indicated that the emission efficiency of Sn-Pb alloyed perovskite QDs is significantly higher than that of CsPbI₃ QDs.*

The extremely low PLQY value of approximately 0.3% in the reported CsSn_{1-x}Pb_xI₃ QDs may result from the increased intrinsic defects that are associated with Sn vacancies (V_{Sn}), which have very low defect formation energies of ~250 meV. These vacancies can thus form easily and produce a high p-type conductivity as well as deep-level defects that act as nonradiative recombination centers, resulting in severe PL degradation. CsSn_{1-x}Pb_xI₃ QDs (x = 0.4-0.8) in the reported works were synthesized with a high Sn content through the conventional hot-injection method, which may

bring more defects. However, the Sn content in the CsSn_{1-x}Pb_xI₃ QDs we prepared only accounts for 9% of the B-site elements, and the main body of the QDs synthesized at room temperature is still CsPbI₃. Thus, the nonradiative recombination centers caused by stannous ions in our QDs are very limited. In addition, a certain amount of stannous ions is also beneficial for passivating lead vacancy defects and enhancing the crystal quality of CsSn_{1-x}Pb_xI₃ QDs. As a result, our CsSn_{1-x}Pb_xI₃ QDs have a higher PLQY as compared with that in the reported works.

5. Line 211-215, The author mentioned that defect passivation through stannous alloying can passivate the uncoordinated Pb atoms in the crystal lattice. Is this correct? I do not see evidence that stannous can coordinate with the uncoordinated Pb atoms. Could the authors give a diagram of crystal structure with the incorporation process and describe how Sn²⁺ can coordinate with Pb²⁺?

Response: *We thank the reviewer for the insightful comments. Sorry for the inaccurate expression. The efficacy of stannous ions is mainly to passivate lead-related defects, which mainly includes two aspects: filling Pb vacancy defects and replacing uncoordinated Pb atoms. The stannous ions do not coordinate with uncoordinated lead atoms. Finally, the corresponding modifications have been made in the revised manuscript to avoid ambiguity and misperception.*

6. Figure S10 and S11 show the absorption and photoluminescence of nanocrystals synthesized at higher temperatures. Clearly, the products are a mixture of 3 to 4 phases/nanostructures. The authors only comment on the primary peak, which is at around 600 nm, but the other peaks are still very sharp (for example, the peaks at 560 nm, 620 nm, 650 nm for Figure S10b, the peaks at 560 nm, 620 nm for Figure S10e, Figure S11b, and the peaks at 620 nm for Figure S11e). What are these peaks? Do these peaks result from Sn-Pb phase segregation or Sn²⁺ - Sn⁴⁺ oxidation? If a mixture

system with unknown composition is photoexcited, then the PL lifetime is meaningless. The authors need to determine the exact composition for each phase and get reliable results from a clean system.

Response: We thank the reviewer for the suggestion. As stated in the second comment from **Reviewer #1**, the broad and multi-peak signals indicate **high size polydispersity** of the resultant NCs. In other words, the multimodal nature of the PL spectra implied **the presence of NCs with different shapes and sizes** in the reaction products and size-dependent effect. By adjusting the reaction parameters, the morphology and size of the product will undergo significant changes, as shown in Fig. R1. The phenomenon of multiple peaks in the PL spectra in any way is not associated with Sn-Pb phase segregation or Sn^{2+} - Sn^{4+} oxidation. In fact, the broad and multi-peak signals are widely present in the studies of lead halide perovskite NCs, as described in the following reported papers (*J. Phys. Chem. Lett.* 2019, 10, 4149-4156; *Chem. Mater.* 2019, 31, 365-375; *Nano Lett.* 2018, 18, 1246-1252; *Adv. Mater.* 2022, 34, 2107105; *Adv. Funct. Mater.* 2022, 32, 2108687).

7. Fig. 3b shows that the pure-iodine Sn-Pb perovskite QDs possess a direct bandgap of 2.12 eV. What is the composition (Sn/Pb ratio) of the system used for DFT calculation here? Is the Sn/Pb ratio for DFT consistent with the experimental composition? How do the authors consider the Sn^{2+} to Sn^{4+} oxidation in the DFT, which is an unavoidable but critical factor for Sn^{2+} perovskites?

Response: We thank the reviewer for the suggestion. The percentage of Sn/(Sn+Pb) in a theoretical model used for DFT calculation is 6.3%; and the actual proportion of Sn/(Sn+Pb) in the experimental product is 9.1%, which is determined by ICP-OES. For the DFT calculation, we did not consider the issue of the Sn^{2+} to Sn^{4+} oxidation, because it is difficult to form Sn^{4+} for Sn-containing perovskites in the computing system, and currently we have not obtained relevant theoretical results. For this issue, we have discussed it with research groups in the field of

theoretical calculation for halide perovskites, such as Yanfa Yan's group from the University of Toledo, and they also have no relevant progress yet due to the same reason. More importantly, the stannous ions in our sample only account for a small portion, and very little Sn²⁺ with the protection of solvent and surface ligands can be oxidized. Therefore, considering the experiment results and the difficulty of the theoretical calculation, we do not consider the issue of Sn²⁺ being oxidized in the DFT calculation at this condition. We will conduct follow-up exploration to determine the real quantity of tetravalent tin in colloidal solution and try to solve the issue by theoretical calculation.

8. In addition to the previous question, I do not see any results characterizing the Sn/Pb ratio in the article, and the entire article does not give the chemical formula of the Sn-Pb material system studied. The authors should give a formula (or Sn/Pb ratio) of the perovskite nanocrystals. Probably extremely low doping of Sn is the reason for the slight change in bandgap: Line 221-224, 'no noticeable change in the exciton absorption and PL emission characteristics of the as-prepared Sn-Pb perovskite QDs with the increase of the Pb/Sn ratio.'

Response: *We thank the reviewer for the suggestion. The actual proportion of Sn/(Sn+Pb) in the yellow emissive sample is 9.1%, which is determined by the quantitative analysis of ICP-OES. The chemical formula of the pure-iodine all-inorganic Sn-Pb perovskite QDs has been given in the revised manuscript. Thus, the incorporation of stannous ions is very limited in pure-iodine all-inorganic Sn-Pb perovskite QDs by this room temperature strongly confined spontaneous crystallization strategy. However, the luminescent properties of perovskite QDs depend entirely on their strong size confinement due to the extremely small QD size, exhibiting a significant blue shift phenomenon; in this case of strong size confinement, the incorporation of a certain amount of*

stannous ions in pure-iodine all-inorganic perovskite QDs shows negligible changes in emission peak positions.

9. Line 291-293, 'QDs with longer emission wavelengths lose their PL performance after being placed in air for one day, and their RT stability is much lower than that of wider bandgap QDs. Therefore, the colloidal stability of the pure-iodine perovskite QDs largely benefits from the small QD size.' The authors missed the information on what the stability is for wider bandgap QDs; please attach a stability test of both systems. The authors mentioned that smaller QDs are more colloiddally stable, but in Line283-285 they mentioned that small QDs have much more traps than larger ones. The authors need to explain why small QDs with more traps (more Cs, Pb or halide vacancies in the structure) show much better stability.

Response: *We thank the reviewer for the insightful comments. The stability comparison in air for the resultant samples with Cs/Pb ratios of 1:4 and 3:2 is displayed in Fig. R5. The experimental result indicates that the room temperature storage stability of small-sized sample is significantly better than that of longer-wavelength sample with larger size. The color of the longer-wavelength sample with Cs/Pb ratio of 3:2 underwent a significant change from deeply red to grayish white, and the PL performance disappeared after being placed in air for one day. This is mainly attributed to the phase transition from metastable photoactive perovskite phase to stable non-perovskite yellow phase. However, the small-sized sample with Cs/Pb ratio of 1:4 still exhibits bright yellow emission after more than ten days in the air.*

Fig. R5 Photographs under daylight (upper) and UV light (bottom) showing the colloidal stability in air for the samples with Cs/Pb ratios of 1:4 and 3:2.

As we all know, perovskite-phase CsPbI_3 is structurally unstable at temperatures below $320\text{ }^\circ\text{C}$ and spontaneously transits into an undesired non-perovskite δ -phase (yellow phase). Because the crystallization rate of halide perovskite NCs is extremely fast, it is very easy for iodine-based reaction system to directly form a stable nonphotoactive yellow phase with a large crystal size through a room temperature process, as shown in Fig. R6. Relatively speaking, larger CsPbI_3 perovskite crystals are more likely to undergo phase transitions, resulting in the disappearance of photoactivity. However, colloidal CsPbI_3 QDs with small crystal size have shown a much more stable perovskite structure due to the strong quantum confinement effect and high surface Gibbs energy from the dominant contribution of large surface/volume ratio (Adv. Mater. 2020, 32, 2002632; J. Mater. Chem. A, 2020, 8, 10226-10232; Science 2016, 354, 92-95). Therefore, in order to obtain a sufficiently stable Sn^{2+} alloyed CsPbI_3 perovskite phase structure at room temperature, reducing the crystal size through room temperature strongly confined strategy is crucial. Although small-sized crystals always show more surface defects compared with the larger-sized crystals, small size is beneficial for maintaining photoactive pure-iodine all-inorganic perovskite structure. Therefore, the crystal size of pure-iodine all-inorganic perovskite QDs plays a dominant role in

room temperature stability for perovskite phase.

Fig. R6 *Photograph under daylight of the resultant product obtained with a high Cs/Pb feed molar ratio of 2:1. It indicates the readily formation of stable non-perovskite δ -phase without photoactive properties through room temperature reaction.*

10. The authors used the model of passivation of Pb vacancy defect to explain the increased stability of Sn-Pb perovskites in Line 328-335. This trend is exactly opposite to the trend of stability and defect states mentioned by the author in the previous question, where larger NCs with less defect show much worse stability. In addition, the Sn^{2+} can easily change into Sn^{4+} even in a glovebox with several ppm of O_2 . The authors need to consider this practical intrinsic instability of Sn either in the simulation or in the discussions. How will this affect the stability of Sn-Pb perovskites?

Response: *We thank the reviewer for the suggestion. According to the response from **Comment 9**, large-sized Sn^{2+} alloyed CsPbI_3 crystals with a higher Cs/Pb feed molar ratio are prone to directly form stable non-perovskite δ -phase without photoactive properties through room temperature reaction. However, colloidal CsPbI_3 QDs with small crystal size have shown a much more stable perovskite structure due to the strong quantum confinement effect and high surface Gibbs energy from the dominant contribution of large surface/volume ratio. Thus, although the large-sized*

pure-iodine all-inorganic crystals exhibit fewer crystal defects, they are more likely to transform into stable non-perovskite δ -phase that have no photoactive properties compared to the small-sized crystals. Overall, the crystal size plays a crucial role in the stability of the pure-iodine all-inorganic perovskite structure, even surpassing the influence of crystal defects.

In this work, the stable sub-5 nm sized pure-iodine all-inorganic Sn-Pb perovskite QDs were synthesized through a direct room temperature strongly confined spontaneous crystallization strategy in a Cs-deficient reaction system without polar solvents. This was mainly due to two aspects: small-size-induced stabilization and Sn-Pb alloying. Based on the same synthesis process with similar crystal sizes, the luminescence properties and stability of pure-iodine all-inorganic Sn-Pb perovskite QDs were significantly improved as compared to pristine CsPbI₃ QDs. The results indicated that the lead-related defects could be passivated after the incorporation of stannous ions. In addition, the actual proportion of Sn/(Sn+Pb) in the yellow emissive sample is 9.1%, which is determined by the quantitative analysis of ICP-OES, indicating very limited incorporation of stannous ions in Sn-Pb perovskite QDs. So the issue of oxidation can be negligible in colloidal solution with the protection of solvent and surface ligands.

11. Line 335-336, 'the formation of vacancy defects gradually becomes difficult with the increased proportion of stannous ions in the Sn-Pb iodide perovskite'. The authors claimed that the increased proportion of Sn can stabilize the structure (better stability than Pb) by decreasing vacancy defects. However, in Line 226-229, the authors mentioned that excessive Sn leads to more defect states. ('excessive incorporation of stannous ions will cause an increase in defect states of Sn-Pb perovskite QDs, leading to intensified nonradiative recombination and PL quenching.'). This seems to be quite contradictory. A detailed relationship between stability, Sn/Pb ratio, vacancy traps, and

Sn⁴⁺ content should be given to support the experimental results.

Response: *We thank the reviewer for the suggestion. The above two aspects mentioned by the reviewer should be viewed separately, as they are not directly related. The statement regarding line 335-336 is for the formation energy theoretical calculations of all-inorganic Sn-Pb iodide perovskite. This theoretical calculation cannot take into account the oxidation of stannous ions, but only theoretically reveals that the formation energy of perovskite structure gradually increases with the increased proportion of stannous ions. This theoretically indicates that the incorporation of stannous ions is beneficial for stability improvement of pure-iodine all-inorganic perovskite structure.*

However, the expression in line 226-229 of the latter corresponds to the experimental results in Fig. S9c. The optical characteristics of pure-iodine all-inorganic Sn-Pb perovskite QDs are dominated by the strong size confinement. Although the exciton absorption and PL emission characteristics show negligible changes with the reduction of the Pb/Sn ratio, the fluorescence lifetime of photogenerated charge carriers is gradually shortened for the PL decay curves (Fig. S9c). Thus, the excessive incorporation of stannous ions will cause an increase in defect states of perovskite QDs, leading to intensified nonradiative recombination. Ultimately though, pure-iodine all-inorganic Sn-Pb perovskite QDs exhibit superior performance as compared with pristine CsPbI₃ QDs. As the Pb/Sn ratio further decreases, the defect states caused by the oxidation of stannous ions will increase to a certain extent. However, there is no significant difference in the colloidal stability of yellow emissive pure-iodine all-inorganic Sn-Pb perovskite QD solution with various Pb/Sn ratios in our study. In addition, the stannous ions in our samples only account for a small portion, and the Sn²⁺ oxidation is very limited with the protection of solvent and surface ligands.

12. Regarding the oxidation from Sn^{2+} to Sn^{4+} , the author only mentioned in Line 169-171 that 'Unfortunately, the purified pure-tin perovskite QD colloidal solution cannot be obtained in open air due to the susceptibility of stannous ions to oxygen gas.'. Sn^{4+} plays a critical role in these nanocrystals synthesized in open air. The authors need to give Pb/Sn ratio and the $\text{Sn}^{2+}/\text{Sn}^{4+}$ ratio from XPS measurements (Figure S8) and ensure they are focusing on incorporating Sn^{2+} as they claimed.

Response: *We thank the reviewer for the suggestion. We attempted to synthesize pure-tin all-inorganic perovskite QDs at room temperature in open air. Although the crude solution of pure-tin all-inorganic perovskite QDs were obtained, it deteriorated quickly and completely during the purification process. In our experiment, the actual proportion of Sn/(Sn+Pb) in the pure-iodine all-inorganic Sn-Pb perovskite QDs is 9.1%, which is determined by the quantitative analysis of ICP-OES.*

In addition, we think the $\text{Sn}^{2+}/\text{Sn}^{4+}$ ratio from XPS measurement is difficult to accurately reflect our experimental results. The entire process of preparing and transferring QD film sample for XPS test in an inert gas atmosphere is not yet achieved due to the limited experimental condition. In our experiment, our QDs in colloidal solutions are protected by solvent and surface ligands and almost not oxidized. For the XPS test, the QDs exposed to air need to be fabricated as the QD film samples on the substrate, they are very prone to oxidation due to the lack of solvent protection. To verify this, the XPS test of QD film samples was carried out in the air, which causes severe oxidation of stannous ions, showing the $\text{Sn}^{2+}/\text{Sn}^{4+}$ ratio of 9:11, as shown in Fig. R7. This result further confirms the successful incorporation of Sn^{2+} in CsPbI_3 perovskite crystal structure. Unfortunately, we are not able to directly measure the proportion of Sn with different valence states in colloidal solution.

Fig. R7 High-resolution Sn 3d XPS spectra of pure-iodine all-inorganic Sn-Pb perovskite QDs. The preparation process of QD film sample for XPS test was carried out in the air, resulting in severe Sn²⁺ oxidation due to the lack of solvent protection.

There are some minor issues:

1. There is another small bump for both Br (at 500 nm) and Cl (at 410 nm) samples. The authors need to make sure Br and Cl samples do not have phase segregation. In addition, the photoluminescence of Br and Cl based Sn-Pb nanocrystals should also be given in Figure S4.

Response: We thank the reviewer for the suggestion. The small bump for both Br (at 500 nm) and Cl (at 410 nm) samples is attributed to the diversity of NC sizes (high size polydispersity), which is consistent with the emission peak position in the PL spectra, as shown in Fig. R8. Finally, the PL spectra of all-inorganic Br- and Cl-based Sn-Pb perovskite QDs have been added in Fig. S4.

Fig. R8 PL spectra of all-inorganic Br- and Cl-based Sn-Pb perovskite QDs. The excitation wavelength for Br-based sample was 365 nm, and 300 nm excitation wavelength was applied for

Cl-based sample.

2. In Figure S8b, the caption (Br 3d) does not match the label in the map (I 3d), please correct this.

Response: *We thank the reviewer for pointing out it. The caption of Figure S8b has been corrected.*

3. All the 'Sn-Pb QDs' should be replaced by the exact chemical formula of the perovskites.

Response: *We thank the reviewer for the suggestion. According to the test results of ICP-OES, the exact chemical formula of the Sn-Pb perovskite QDs has been given in the revised manuscript.*

REVIEWER COMMENTS

Reviewer #1 (Remarks to the Author):

The authors have appropriately addressed my previous comments. In particular, there is now a clear quantitative analysis which sheds important light into the final composition (especially tin/lead ratio). I have no further comments.

Reviewer #2 (Remarks to the Author):

All the issues have been addressed. The paper can be accepted.

Reviewer #3 (Remarks to the Author):

In this manuscript, the authors present their findings regarding the synthesis of a specific nanocrystal composition, $\text{CsSn}_{0.09}\text{Pb}_{0.91}\text{I}_3$, conducted at room temperature in an open-air environment. Their XPS analysis reveals that more than 70% of the tin (Sn) ions exist in the tetravalent Sn^{4+} oxidation state. However, despite the revisions made to the manuscript, my concerns about the potential oxidation of Sn during the open-air synthesis process remain unaddressed. Furthermore, I find that the synthesis method lacks substantial innovation, which raises questions about its overall contribution to the field. Consequently, I am not inclined to recommend the publication of this manuscript in Nature Communications.

The following are detailed comments on the manuscript:

(1) The central aspect of this work is the synthetic process. Nevertheless, it bears a striking resemblance to a prior publication (Nanoscale, 2021, 13, 4899-4910) in which uniform cubic-phase CsPbI_3 nanocrystals were synthesized at room temperature in an open-air setting, with a photoluminescence peak at 569 nm. It appears that this manuscript could be perceived as a follow-up to the earlier work, with the distinction of achieving synthesis for only one specific composition, $\text{CsSn}_{0.09}\text{Pb}_{0.91}\text{I}_3$, containing 9% Sn. This limited scope considerably diminishes the potential impact of the method.

(2) In terms of size control, it's noteworthy that other studies have demonstrated the synthesis of ultrasmall Sn-Pb dots (Sn content from 0% to 100%) measuring less than 3 nm (ACS Energy Lett. 2017, 2, 1190-1196, Chem. Mater. 2020, 32, 1089-1100, Light Sci. Appl. 2023, 12, 208). As a result, the novelty and scientific connotation of this method is limited.

(3) With regard to composition, the authors indicate that only one specific component (or a very limited range of Sn doping below 10%) can be synthesized using their method. Additionally, XPS measurements reveal that approximately 70% of the Sn atoms exist in the Sn^{4+} oxidation state, while 30% are in the Sn^{2+} state (as evidenced by the twice-stronger Sn^{4+} peak). However, the manuscript lacks sufficient evidence to substantiate the claim that nanocrystals are not oxidized during the open-air synthesis process.

(4) If the authors think that the quantum dots become oxidized upon loading into the XPS chamber, which contradicts their initial claim of stability after more than one day in open air, it is essential that they determine the $\text{Sn}^{2+}/\text{Sn}^{4+}$ ratio for the nanocrystals synthesized in open air (clarifying what materials they are actually producing in an open-air environment). At present, the manuscript makes assumptions without providing any concrete evidence, positing that nanocrystals synthesized in an

open-air environment contain 100% Sn²⁺ and 0% Sn⁴⁺, yet they exhibit instability (to form 70% Sn⁴⁺) when introduced to the XPS chamber. I am not convinced by this argument.

Point-by-point responses to the reviewers

We are very grateful to Reviewers #1 and #2 for their positive feedbacks on this work and greatly appreciate Reviewer #3 for the additional much-valued comments. We have carefully considered the comments from Reviewer #3 and revised the paper accordingly. We have incorporated all the suggestions and addressed all the comments in our revised manuscript. The main corrections in the revised manuscript and Supplementary Information are marked in red, and a comprehensive point-by-point response to the reviewers is given below.

Response to Reviewer #1 :

The authors have appropriately addressed my previous comments. In particular, there is now a clear quantitative analysis which sheds important light into the final composition (especially tin/lead ratio). I have no further comments.

Response: *We sincerely thank the reviewer for thorough and careful reading of our manuscript, and appreciate the reviewer's recognition of our work, which encourage us to continue engaging in cutting-edge research.*

Response to Reviewer #2 :

All the issues have been addressed. The paper can be accepted.

Response: *We sincerely thank the reviewer for the acceptance of our paper, which encourage us to continue engaging in cutting-edge research.*

Response to Reviewer #3 :

In this manuscript, the authors present their findings regarding the synthesis of a specific

nanocrystal composition, $\text{CsSn}_{0.09}\text{Pb}_{0.91}\text{I}_3$, conducted at room temperature in an open-air environment. Their XPS analysis reveals that more than 70% of the tin (Sn) ions exist in the tetravalent Sn^{4+} oxidation state. However, despite the revisions made to the manuscript, my concerns about the potential oxidation of Sn during the open-air synthesis process remain unaddressed. Furthermore, I find that the synthesis method lacks substantial innovation, which raises questions about its overall contribution to the field. Consequently, I am not inclined to recommend the publication of this manuscript in Nature Communications.

Response: *We sincerely thank the reviewer for the thorough and careful reading of our manuscript and appreciate the reviewer for the comments.*

Firstly, it should be pointed out that we cannot directly conduct XPS testing on quantum dot (QD) colloidal solutions to obtain accurate oxidation ratio of stannous (Sn^{2+}) ions in the solution. However, Sn^{2+} oxidation in QD thin film samples for XPS measurement exposed to air are very serious due to the large surface area caused by their small size. Secondly, we do not deny the existence of oxidation of Sn^{2+} ions during the synthesis process, but we can speculate that the oxidation of stannous ions in the colloidal solution is relatively mild with the protection of solvent and surface ligands. Moreover, the tin ions in our samples only account for a small portion below 10%. Thirdly, the $\text{CsSn}_{0.09}\text{Pb}_{0.91}\text{I}_3$ QDs exhibited the enhanced PL intensity, prolonged fluorescence lifetime and improved colloidal stability as compared with the pristine CsPbI_3 QDs, which is mainly attributed to the incorporation of Sn^{2+} ions. If Sn^{2+} ions undergo severe oxidation in colloidal solutions, a large number of tin vacancy defects will inevitably be generated, exacerbating nonradiative recombination and weakening luminescence performance. Therefore, the Sn^{2+} oxidation during colloid synthesis is not severe due to the enhanced PL performance for $\text{CsSn}_{0.09}\text{Pb}_{0.91}\text{I}_3$ QDs.

Taking into account the above three points, the Sn²⁺ oxidation is very limited during the open-air synthesis process, but we cannot directly characterize the oxidation ratio of Sn²⁺ in the colloidal solution through XPS testing. To preliminarily evaluate the valence state of Sn ions in the synthesized colloidal QDs, we prepared XPS film sample in a glovebox with inert atmosphere. The measured result will be detailed later.

As for the innovation aspect of the synthesis method, please refer to the responses to Comments (1) and (2).

The following are detailed comments on the manuscript:

(1) The central aspect of this work is the synthetic process. Nevertheless, it bears a striking resemblance to a prior publication (Nanoscale, 2021, 13, 4899-4910) in which uniform cubic-phase CsPbI₃ nanocrystals were synthesized at room temperature in an open-air setting, with a photoluminescence peak at 569 nm. It appears that this manuscript could be perceived as a follow-up to the earlier work, with the distinction of achieving synthesis for only one specific composition, CsSn_{0.09}Pb_{0.91}I₃, containing 9% Sn. This limited scope considerably diminishes the potential impact of the method.

Response: *We thank the reviewer for the comment. The first thing to declare is that apart from the synthesis conditions (at room temperature in open air), this work is completely different from Nanoscale paper. As shown in Table R1, the chemical reagents used in our work are almost different from those in Nanoscale paper, the synthesis of our work is a new room temperature reaction system for all-inorganic Cs-based pure-iodine Sn-Pb mixed perovskite QDs. To further demonstrate the importance of the reaction system, here, we give an example: the same hot-injection method was used in both works (Nano Lett. 2015, 15, 6, 3692-3696 and J. Am. Chem. Soc. 2018, 140, 7,*

2656-266), but the chemical reagents used were different. The latter with a new reaction system also had sufficient novelty and was published in a high-level journal. More importantly, the preparation of metal precursor in Nanoscale paper requires a Schlenk-line equipment with heating and inert atmosphere, which is relatively complex and inconvenient as compared with our preparation process. So our preparation method has clear superiority. The research object in Nanoscale paper is pristine cesium lead halide CsPbX_3 perovskite nanocrystals (NCs), and the direct room temperature preparation of pure-iodine Cs-based Sn-Pb alloyed perovskite QDs is not involved. However, the main task of this work is to develop a superior room temperature synthesis pathway for pure-iodine Sn-Pb alloyed perovskite QDs, with the assistance of size-stabilized perovskite phase and Sn/Pb direct alloying. This work has pioneered the direct room temperature preparation of pure-iodine all-inorganic Sn-Pb alloyed perovskite QDs, and provided a new pathway for the room temperature synthesis of pure-iodine all-inorganic perovskite QDs with size-stabilized strategy. Therefore, we believe that our work is very different from the Nanoscale paper. Additionally, our work is not limited to only one specific composition ($\text{CsSn}_{0.09}\text{Pb}_{0.91}\text{I}_3$), the final Sn content in QDs can be controlled within a certain range by adjusting the ratio of Sn/Pb precursors. This strongly confined room temperature approach was universal for wider bandgap bromine- and chlorine-based all-inorganic and iodine-based hybrid perovskite QDs.

Table R1. Comparisons of synthesis technology for Nanoscale paper and our study.

Perovskite	Synthetic method	Reactants	Emission color	Ref.	Remarks
CsPbI_3	Room temperature synthesis with relatively complex preparation processes of precursor solutions	Cs_2CO_3 , PbO , triphenylphosphine (Ph_3P), oleic acid (OA), oleylamine (OAm), stearic acid, hexadecylamine, mesitylene, solid iodine (I_2), chloroform (CH_2Cl), dichloromethane (CH_2Cl_2), toluene	Orange-red	Nanoscale, 2021, 13, 4899-4910	Need to use Schlenk-line equipment to prepare metal precursor, and not pure-iodine all-inorganic Sn-Pb perovskite QDs

$\text{CsSn}_{0.09}\text{Pb}_{0.91}\text{I}_3$	Room temperature synthesis with a new reaction system	Cesium acetate, lead acetate, tin(II) 2-ethylhexanoate, toluene, OA, OAm, iodotrimethylsilane	Yellow	This work	An efficient and facile room temperature method, all-inorganic pure-iodine Sn-Pb perovskite QDs
--	--	---	--------	------------------	--

(2) In terms of size control, it's noteworthy that other studies have demonstrated the synthesis of ultrasmall Sn-Pb dots (Sn content from 0% to 100%) measuring less than 3 nm (ACS Energy Lett. 2017, 2, 1190-1196, Chem. Mater. 2020, 32, 1089-1100, Light Sci. Appl. 2023, 12, 208). As a result, the novelty and scientific connotation of this method is limited.

Response: We thank the reviewer for the comment. Here we need to emphasize that the room temperature synthesis of pure-iodine all-inorganic perovskite colloidal QDs still faces significant difficulties due to the phase instability and smaller ionic size of Cs^+ ions. Besides, room temperature synthesis without inert gas protection has attracted more attention due to its easy manipulation and mild reaction conditions. This work proposes **a new room temperature strategy with strongly confined synthesis for yellow emissive pure-iodine all-inorganic Sn-Pb perovskite colloidal QDs with high crystal quality, which has never been reported before.** As is well known, it is extremely easy to synthesize iodine-based perovskite colloidal QDs through hot-injection method. It is worth noting that the studies mentioned by the reviewer all involve hot-injection route rather than room temperature synthesis (see Table R2). Even more remarkably, room temperature synthesis of pure-iodine Cs-based perovskite QDs is more difficult compared to organic-inorganic hybrid counterparts, because the smaller ionic size of Cs^+ is not enough to maintain the lead-iodine octahedron to form a periodic perovskite structure. **Therefore, these mentioned works are completely unrelated to our work.** In other words, it does not make much sense to compare them. Moreover, the size control of the product NCs at room temperature in our work is realized by

adjusting the ratio of Cs/Pb precursors. It is also completely different from the regulation pathway of hot-injection synthesis in the mentioned studies.

Table R2. Comparisons of synthesis technology for pure-iodine perovskite QDs in the mentioned references and our study.

Perovskite	Synthetic method	Reactants	Ref.	Remarks
$CH_3NH_3SnI_3$ $CH_3NH_3Sn_xPb_{1-x}I_3$	Hot-injection route + Cation exchange (70 °C)	SnI_2 , trioctylphosphine (TOP), octadecene, OA, OAm, methylamine, tetrahydrofuran, PbI_2	ACS Energy Lett. 2017, 2, 1190-1196	Not a room temperature method, and not all-inorganic perovskite QDs
$CsPbI_3$ $CsSn_xPb_{1-x}I_3$	Hot-injection route	PbI_2 , trioctylphosphine oxide (TOPO), Cs_2CO_3 , OA, OAm, octadecene, SnI_2	Chem. Mater. 2020, 32, 1089-1100	Not a room temperature method
$CsSn_xPb_{1-x}I_3$	Hot-injection route	Cs_2CO_3 , OA, OAm, octadecene, PbI_2 , SnI_2 , TOP	Light Sci. Appl. 2023, 12, 208	Not a room temperature method
$CsSn_{0.09}Pb_{0.91}I_3$	Room temperature synthesis	Cesium acetate, lead acetate, tin(II) 2-ethylhexanoate, toluene, OA, OAm, iodotrimethylsilane	This work	An efficient and facile room temperature method, all-inorganic Sn-Pb perovskite QDs

(3) With regard to composition, the authors indicate that only one specific component (or a very limited range of Sn doping below 10%) can be synthesized using their method. Additionally, XPS measurements reveal that approximately 70% of the Sn atoms exist in the Sn^{4+} oxidation state, while 30% are in the Sn^{2+} state (as evidenced by the twice-stronger Sn^{4+} peak). However, the manuscript lacks sufficient evidence to substantiate the claim that nanocrystals are not oxidized during the open-air synthesis process.

Response: We thank the reviewer for the comment. In this work, the successful synthesis of high-quality yellow emissive pure-iodine all-inorganic Sn-Pb perovskite QDs is attributed to the strongly confined size-stabilized strategy and the incorporation of stannous ions. The final Sn content in QDs can be controlled within a certain range by adjusting the ratio of Sn/Pb precursors. However, the limited amount of Sn incorporation may be due to the difficulty in alloying caused by

the small QD size. For the XPS test, the small QDs exposed to air need to be fabricated as the QD film samples on the substrate, they are very prone to oxidation due to the lack of solvent protection and larger surface area. As a result, there were more Sn^{4+} ions in the test result. **To further confirm this, we prepared XPS film sample in a glovebox with inert atmosphere.** As shown in Fig. R1, the test result indicates that **the fitting peaks at 495.0 and 486.5 eV are attributed to the energy levels of $3d_{3/2}$ and $3d_{5/2}$ for Sn^{2+} , respectively, and no obvious Sn^{4+} signal is detected in the testing sample.** Therefore, it can be confirmed that the oxidation of Sn^{2+} ions in the QD colloidal solution is not significant during the synthesis process. Additionally, we do not deny the existence of oxidation of Sn^{2+} ions during the synthesis process, but we can speculate that the oxidation of Sn^{2+} ions in the colloidal solution is relatively mild with the protection of solvent and surface ligands.

Fig. R1 High-resolution Sn 3d XPS spectra of $\text{CsSn}_{0.09}\text{Pb}_{0.91}\text{I}_3$ QDs. The preparation process of QD film sample for XPS test was carried out in a glovebox with inert atmosphere.

(4) If the authors think that the quantum dots become oxidized upon loading into the XPS chamber, which contradicts their initial claim of stability after more than one day in open air, it is essential that they determine the $\text{Sn}^{2+}/\text{Sn}^{4+}$ ratio for the nanocrystals synthesized in open air (clarifying what materials they are actually producing in an open-air environment). At present, the manuscript makes

assumptions without providing any concrete evidence, positing that nanocrystals synthesized in an open-air environment contain 100% Sn^{2+} and 0% Sn^{4+} , yet they exhibit instability (to form 70% Sn^{4+}) when introduced to the XPS chamber. I am not convinced by this argument.

Response: *We thank the reviewer for the comment. For XPS testing, the entire process of preparing and transferring QD film sample was carried out in the air, leading to severe oxidation of Sn^{2+} in QDs due to the lack of solvent protection and larger surface area. However, when the pure-iodine all-inorganic Sn-Pb perovskite QDs exist in the form of colloidal solutions, the oxidation of Sn^{2+} is not very significant due to the protection of solvent and surface ligands. To further demonstrate the valence state of Sn ions in the synthesized colloidal QDs, we prepared XPS film sample in a glovebox with inert atmosphere. As shown in Fig. R1, no obvious Sn^{4+} signal was detected in the testing sample, except for the Sn^{2+} peaks. Therefore, the as-synthesized product QDs are composed of Sn^{2+} ions, and it cannot be ruled out that there may be a small amount of Sn^{2+} oxidation. It's important to reiterate that the Sn^{2+} ions in QD film sample exposed to air is unstable due to susceptibility to oxygen gas, and the QD colloidal solution is relatively stable due to the protection of solvent and surface ligands.*

REVIEWERS' COMMENTS

Reviewer #3 (Remarks to the Author):

The authors have appropriately addressed all my concerns. However, the current abstract may be misleading, as readers might infer that sub-5nm Sn/Pb QDs with yellow emission have not been reported before. Please modify the wording and incorporate the following references into the main text:

1. Nanoscale, 2021, 13, 4899-4910 (Room-temperature open-air synthesis of Sn/Pb QDs with yellow emission)
2. ACS Energy Letters 2017, 2, 1190-1196, Chem. Mater. 2020, 32, 1089-1100, Light Science & Applications 2023, 12, 208 (sub-5nm Sn/Pb QDs).

Point-by-point responses to the reviewers

Reviewer #3 (Remarks to the Author):

The authors have appropriately addressed all my concerns. However, the current abstract may be misleading, as readers might infer that sub-5nm Sn/Pb QDs with yellow emission have not been reported before. Please modify the wording and incorporate the following references into the main text:

1. Nanoscale, 2021, 13, 4899-4910 (Room-temperature open-air synthesis of Sn/Pb QDs with yellow emission)
2. ACS Energy Letters 2017, 2, 1190-1196, Chem. Mater. 2020, 32, 1089-1100, Light Science & Applications 2023, 12, 208 (sub-5nm Sn/Pb QDs)

Response: *We sincerely thank the reviewer for thorough and careful reading of our manuscript, and appreciate the reviewer's recognition of our work, which encourage us to continue engaging in cutting-edge research. Additionally, we have made appropriate modifications according to the comment from the reviewer.*